# Self-organizing layers from complex molecular anions

Jonas Warneke [1], Martin E. McBriarty [1], Shawn L. Riechers [1], Swarup China[2], Mark H. Engelhard [2], Edoardo Aprà [2], Robert P. Young [2], Nancy M. Washton [2], Carsten Jenne [3], Grant E. Johnson[1] & Julia Laskin[1,4]

The formation of traditional ionic materials occurs principally via joint accumulation of both anions and cations. Herein, we describe a previously unreported phenomenon by which macroscopic liquid-like thin layers with tunable self-organization properties form through accumulation of stable complex ions of one polarity on surfaces. Using a series of highly stable molecular anions we demonstrate a strong influence of the internal charge distribution of the molecular ions, which is usually shielded by counterions, on the properties of the layers. Detailed characterization reveals that the intrinsically unstable layers of anions on surfaces are stabilized by simultaneous accumulation of neutral molecules from the background environment. Different phases, self-organization mechanisms and optical properties are observed depending on the molecular properties of the deposited anions, the underlying surface and the coadsorbed neutral molecules. This demonstrates rational control of the macroscopic properties (morphology and size of the formed structures) of the newly discovered anion-based layers.

[1] Physical Sciences Division, Pacific Northwest National Laboratory, 902 Battelle Boulevard, P.O. Box 999, MSIN K8-88, Richland, WA 99352, USA. [2] Environmental Molecular Sciences Laboratory, Pacific Northwest National Laboratory, P.O. Box 999, Richland, WA 99352, USA. [3] Fakultät für Mathematik und Naturwissenschaften, Anorganische Chemie, Bergische Universität Wuppertal, Gaußstraße 20, Wuppertal 42119, Germany. [4] Department of Chemistry, Purdue University, West Lafayette, IN 47907, USA. Correspondence and requests for materials should be addressed to J.W. (email: Jonas.Warneke@gmail.com) or to J.L. (email: jlaskin@purdue.edu)

Precise control of the composition and self-organization of layers on surfaces is key to the rational design of functional materials[1] with broad applications in microelectronics[2], sensing[3], microfluidics[4], and catalysis[4]. Deposition from the gas phase is a promising way to achieve controlled layer-by-layer deposition and avoid detrimental contamination[5]. Self-organization of material is usually achieved by fine tuning the relatively weak intermolecular interactions[6,7] that define the morphology of structures on surfaces from the nanoscale up to the microscale and macroscale[8]. In contrast, substantially stronger forces control interactions of charged species, which have been shown, for example, to determine the layer structure of electron donating organic molecules on metal surfaces[9,10]. The use of ions as functional building blocks for self-organizing layers is usually challenging, because counterions are inevitably present in the condensed phase, often resulting in the formation of highly stable solids. Deposition of ions without their counterions from the gas phase is enabled by ion soft landing, a technique first introduced by Cooks and coworkers[11], which enables immobilization of ions of one polarity on a support. Soft landing has been used previously for conformational enrichment of peptides[12], understanding protein folding on surfaces[13,14], deposition of intact protein assemblies for microscopic imaging[15], generation of microarrays for biological screening[16], preparation of well-defined model catalysts through deposition of ionic clusters, nanoparticles and organometallics[17–23] and in material sciences (e.g., thin composite materials[24] and processing of graphene[25]). In these studies, relatively low coverage samples were prepared for characterization and surface science experiments[26]. Recently, brighter ionization sources with the potential to produce macroscopic quantities of condensed-phase material on surfaces have become available[27–31] and have been used, for example, to generate crystalline phases[30]. While neutralization of ions occurs for many deposited species, retention of charge by deposited ions also has been discussed previously[32,33] and may open the way to prepare materials with unique properties. We previously showed that cations retain their charge when deposited at low coverages on thin insulating layers such as a self-assembled monolayer (SAM) on top of a conductive metal surface[34,35]. Image charges induced by the soft landed ions in the underlying conductive metal substrate are responsible for balancing the overall charge at the interface, which may be rationalized using a parallel plate capacitor model (see Supplementary Note 1 and Supplementary Fig. 1 for details)[36]. Neutralization of cations occurs at higher coverages where electron transfer through the insulating layer becomes possible due to buildup of a potential on the surface[35]. Surface potentials also have been shown to drive ion transfer in certain materials[37,38]. In contrast, our recent studies of stable polyoxometalate Keggin anions, $PM_{12}O_{40}^{3-}$ (M=Mo, W), pointed strongly to preservation of their negative charges after deposition[39,40]. In addition, we showed that deposition of these anions without their countercations onto carbon nanotube electrodes enhanced the capacitance and stability of a macroscopic supercapacitor device[41], emphasizing the technological relevance of such layers and the need to better understand their properties.

We combined a dual ion funnel vacuum interface[27] with our soft landing apparatus[42] which substantially increased the flux of mass-selected ions delivered to surfaces. Deposited layers, visible with the naked eye, now appear on surfaces within hours. Here we show that mass-selected gaseous anions, prepared in a mass spectrometer, may accumulate on surfaces to an extent that condensed-phase liquid-like layers are formed. We further show that the formation of these layers is based on the accumulation of neutral molecules from the gas phase into the assemblies of deposited anions. These layers exhibit self-organization behavior

that may be sensitively tuned by the properties of the mass-selected anions. This unexpected finding opens up unique opportunities for controlling self-organization of materials obtained by deposition of complex anions. In order to understand the effect of the anion on the macroscopic properties of the layers, we employ a family of stable anions with similar structures but different internal charge distributions. Specifically, electronically and chemically stable dianionic halogenated dodecaborates $[B_{12}X_{12}]^{2-}$ (X=F, Cl, Br, I) that have broad applications in catalysis and synthesis[43,44] were chosen as a model system, because their core-shell charge distribution changes systematically with the halogen (X). We characterize the self-organization of the deposited layers after exposure to ambient conditions and observe clear differences which are correlated to the internal charge distributions of the soft landed anions, surface properties and type of accumulated neutral molecules from the gas phase. The self-organization properties of the layers introduced here can be sensitively controlled by these parameters.

## Results

**Structure of investigation**. The presentation of the results is structured as follows. First, we describe the preparation, macroscopic behavior, and chemical characterization of the molecular components of $[B_{12}Cl_{12}]^{2-}$-based layers. Next, we present the influence of several key parameters on the macroscopic properties of the layers by systematically varying the halogen ligand of the anion, the environment, surface properties, and co-adsorbed molecules from the gas phase.

**Preparation of anion-based layers**. Approximately $1 \times 10^{15}$ ions ($\approx 0.05\,\mu mol$) of $[B_{12}Cl_{12}]^{2-}$ were deposited onto 3 mm diameter circular spots on fluorinated SAMs (FSAM, 1H,1H,2H,2H-Perfluorodecanethiol) on gold-coated Si via soft landing of mass-selected ions under vacuum ($6 \times 10^{-5}$ Torr) with stable currents of ~3 nA. FSAMs were selected as deposition substrates due to their previously demonstrated ability to retain the charge of deposited cations and anions [32,40].

**Macroscopic layer behavior**. When a $[B_{12}Cl_{12}]^{2-}$-based layer was exposed to ambient conditions, a change visible to the naked eye occurred in the time frame of minutes. These changes were examined using dark field optical microscopy (Fig. 1). The deposited spot, visible in bright field images (see Supplementary Note 2 and Supplementary Fig. 2), was not initially observed in dark field, indicating that no features were present that scatter light. Following exposure to air, the borders of the layer became visible in the dark field images (Fig. 1a). Circular holes in the layers emerged and grew in diameter over time (Fig. 1b). The borders then merged into each other to form rims, which subsequently ruptured and generated droplets (Fig. 1c) arranged in hexagonal shapes. The size of the self-organized structures was shown to be controlled by varying the amount of deposited material (see images for $10^{13}$, $10^{14}$, and $10^{15}$ ions in Supplementary Note 4 and Supplementary Fig. 6). In contrast, the deposition rate did not influence the result. This time-dependent evolution of the morphology of the deposited layers, known as dewetting[45], has been extensively investigated as a tool for formation of defined structures for technological applications including nanoprinting[46], fabrication of microelectromechanical devices[47,48], and sensor arrays[4,49]. To the best of our knowledge, the dewetting behavior shown in Fig. 1 has not been previously observed for layers prepared by soft landing of complex anions. The process was also examined using atomic force microscopy (AFM) (see Supplementary Note 3, Supplementary Figs. 3–5). The results suggest that the thickness near the middle of the layer

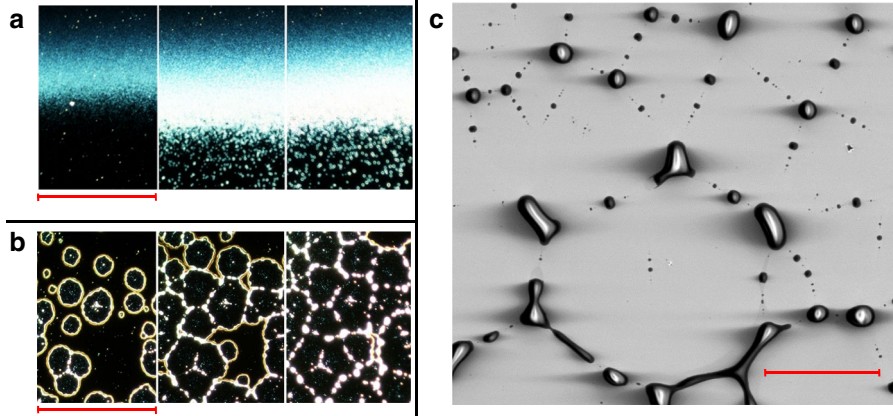

**Fig. 1** Development of a $[B_{12}Cl_{12}]^{2-}$-based layer under ambient conditions. Left: Dark field optical images showing the development of **a** the border (left to right 2, 30, 60 min, scale bar=435 μm) and **b** the middle of the spot (left to right 1, 8, 15 h, scale bar=140 μm) generated by mass-selected deposition of $[B_{12}Cl_{12}]^{2-}$ onto FSAM over time during exposure to ambient conditions. Dewetting is observed throughout the layer. **c** Scanning electron microscopy (SEM) image showing the formation of droplets at the end of the dewetting process (scale bar=100 μm)

**Table 1 XPS elemental composition of the $[B_{12}Cl_{12}]^{2-}$ based layer after deposition of $3 \times 10^{15}$ ions**

| Element | Atomic % |
|---------|----------|
| C (1s)  | 59.0 |
| Cl (2p) | 14.0 |
| B (1s)  | 12.6 |
| O (1s)  | 11.3 |
| N (1s)  | 2.4 |
| Si (2p) | 0.8 |

The electron configuration that corresponds to the detected signal is given in parentheses

in the initial stage of dewetting is roughly 55 nm and 145 nm after soft-landing of $9 \times 10^{14}$ ions and $3 \times 10^{15}$ ions, respectively.

**Chemical analysis of $[B_{12}Cl_{12}]^{2-}$-based layers**. The chemical composition of a $[B_{12}Cl_{12}]^{2-}$-based layer was examined in situ during deposition using infrared reflection-absorption spectroscopy (IRRAS)[42] (Fig. 2a). All observed IR signals grew continuously during ion deposition proportional to the amount of deposited ions (see supplementary Note 5, supplementary Fig. 7). The IR spectrum contained the B–Cl stretching vibration at 1030 cm$^{-1}$ characteristic of the doubly charged $[B_{12}Cl_{12}]^{2-}$[50]. Experimental[51] and calculated IR spectra of neutral halogenated dodecaborates contain additional IR signals originating from a Jahn-Teller distortion (see Supplementary Note 5 and Supplementary Figs. 8 and 9). The absence of these bands in the experimental IR spectrum suggests that the anions retain their charge on the FSAM. Additional IR bands were attributed to adventitious hydrocarbons that accumulated on the charged surface during ion deposition (see Supplementary Note 5 and Supplementary Fig. 10). These bands were only observed in conjunction with the anion deposition and grew in proportion with the B-Cl band. We propose that this process is driven by the presence of a large number of ions on the surface. This assertion is supported by the lack of hydrocarbon accumulation on bare SAMs residing for many hours in the vacuum system. Furthermore, this effect was not observed for soft landed cations, which are more prone to charge reduction on the surface. We note that no significant water bands were detected and the results were independent of the solvent used in the ESI process, indicating that small molecules present in the instrument background are not incorporated into the layer. The layers were also characterized ex-

situ by nanospray desorption electrospray ionization mass spectrometry (nano-DESI MS)[52]. Nano-DESI enables the detection of the accumulated organic species observed as sodium adducts in positive ion mode (Fig. 2b). Based on the accurate mass assignment and fragmentation spectra (see Supplementary Note 6, Supplementary Table 1 and Supplementary Fig. 11, 12), the co-adsorbed molecules were identified to be predominantly phthalates with different chain length, which are typical plasticizers used in rubber components of the vacuum apparatus[53]. Intact $[B_{12}Cl_{12}]^{2-}$ is the dominant peak observed in negative ion mode (Fig. 2c). X-ray photoelectron spectroscopy (XPS) measurements on layers held under nitrogen confirmed that water did not accumulate in substantial amounts in the as-prepared layers based on the amount of oxygen-bound carbon and carbon-to-oxygen ratio (see Table 1 and Supplementary Note 7, Supplementary Fig. 13, 14). In contrast, XPS after dewetting points to a substantial increase of the oxygen content (Supplementary Table 2). The elemental composition in Table 1 reveals that roughly 60 carbon atoms are accumulated per deposited $[B_{12}Cl_{12}]^{2-}$ ion. This suggests that approximately two organic molecules are accumulated per deposited anion based on the molecular formulas of the hydrocarbons. Furthermore, the deposited material was extracted from five high-coverage samples, combined into 600 μL of deuterated methanol, and analyzed by nuclear magnetic resonance (NMR) spectroscopy. The $^{11}B$ NMR spectra (Fig. 2d) are consistent with $[B_{12}Cl_{12}]^{2-}$ (see also Supplementary Note 8 and Supplementary Figs. 15, 16). Collectively, these results provide strong evidence that (1) most of the deposited anions stay intact as charged species on the FSAM, and (2) the layers are stabilized by co-adsorption of organic molecules, mainly phthalates, from the gas phase background of the vacuum chamber. For further analytical details, please see Supplementary Note 5-9.

**Influence of anion properties**. Layers generated using mass-selected deposition of $[B_{12}X_{12}]^{2-}$ (X=F, Cl, Br, I) anions under identical conditions contained similar amounts of hydrocarbons (see Supplementary Note 7 and Supplementary Fig. 10) and appear optically similar under vacuum. However, striking differences were observed in the evolution of the macroscopic layers over time under ambient conditions (Fig. 3). Specifically, the $[B_{12}Br_{12}]^{2-}$ and $[B_{12}I_{12}]^{2-}$-based layers were stable, and no immediate visible changes occurred when they were exposed to ambient conditions. In contrast, soft landing of $[B_{12}F_{12}]^{2-}$

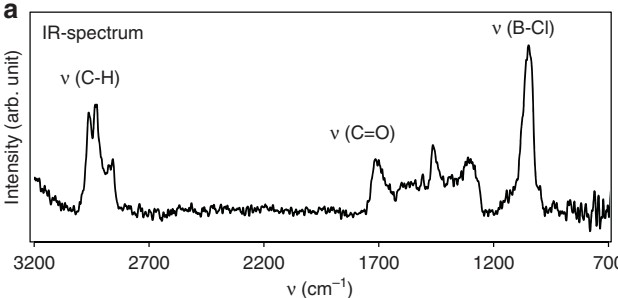

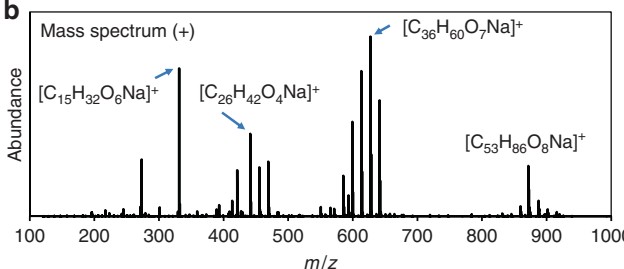

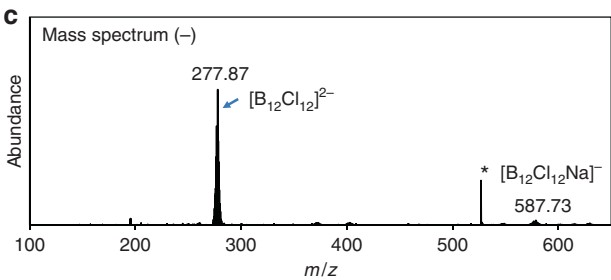

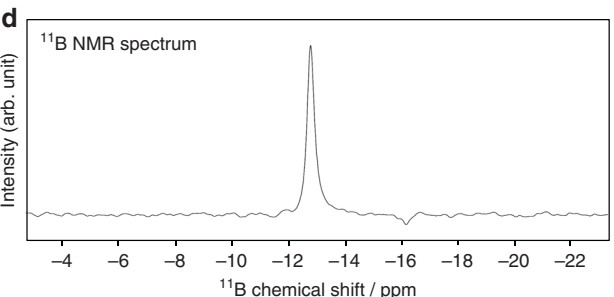

**Fig. 2** Chemical analysis of the layers produced by soft landing of $[B_{12}Cl_{12}]^{2-}$ onto FSAM. **a** In-situ IR spectrum acquired during ion deposition, Ex-situ analysis: **b** positive and **c** negative mode nano-DESI MS spectra of the deposited spot showing a distribution of hydrocarbons, and the intact deposited anions, respectively. The peak marked with a star was identified as solvent signal, **d** $^{11}B$ NMR signal detected from soft-landed $[B_{12}Cl_{12}]^{2-}$ dissolved in methanol-d4. The chemical shift of $[B_{12}Cl_{12}]^{2-}$ was −13.48 ppm as referenced to an external sample of 189 mM boric acid in $D_2O$

produced layers that evolved in air much more rapidly (in seconds) than layers containing $[B_{12}Cl_{12}]^{2-}$ (the smooth clear layer of $[B_{12}F_{12}]^{2-}$ observed through a viewport under vacuum became immediately murky upon exposure to air). The dewetting mechanisms were also different for the $[B_{12}F_{12}]^{2-}$ and $[B_{12}Cl_{12}]^{2-}$-based layers. Specifically, in the thickest region of the high coverage $[B_{12}Cl_{12}]^{2-}$-based layers, dust particles trapped inside the layer served as initiation points. Optical images showed these dust particles in the center of almost each growing circle (Fig. 3). Dewetting theory classifies this process as nucleation growth and the corresponding layers as metastable[54]. In contrast, formation of holes in the $[B_{12}F_{12}]^{2-}$ based layers occurred with considerably higher abundance everywhere in the layer, resulting

in a larger number of smaller sized structures compared to $[B_{12}Cl_{12}]^{2-}$. No defect centers were necessary for formation of free surface areas, suggesting that $[B_{12}F_{12}]^{2-}$-based layers were less stable than the $[B_{12}Cl_{12}]^{2-}$-based layer[54]. Liquid-like dewetting was also observed for $[B_{12}F_{12}]^{2-}$-based layers. However we observed a grainy morphology of the $[B_{12}F_{12}]^{2-}$-based layer and a solid-like morphology of the dewetted droplets distinctly different from the smooth liquid-like layers and droplets formed by the other anion-based layers (Fig. 3, and Supplementary Note 10, Supplementary Figs. 17-19).

**Influence of environmental conditions.** Removal of the surfaces from vacuum and exposure to ambient conditions changes the pressure in the gas phase above the layers which may affect contact angles[55] and, therefore, the likelihood of dewetting. However, controlled venting of the deposition instrument with $N_2$ and storage of a $[B_{12}Cl_{12}]^{2-}$ layer under nitrogen at atmospheric pressure did not result in dewetting. Moisture in air was found to drive the dewetting process. Fast dewetting (<1 min) of the $[B_{12}Cl_{12}]^{2-}$ layer was induced by blowing wet nitrogen over the surface which produced droplet patterns equivalent to those shown in Figs. 2 and 3. Consistent with this observed strong influence of moisture on the dewetting process, we observed a substantial increase in the oxygen-to-boron ratio after dewetting of the layer as measured using XPS (compare Table 1 and supplementary Table 2). We estimate the intake of roughly 20–30 water molecules per deposited anion in the dewettwd layer (compare Table 1 with Supplementary Table 2). This accelerated dewetting could be interrupted at any stage by removal of the flow of moist gas. Furthermore, we found that the holes in $[B_{12}Cl_{12}]^{2-}$-based layers stopped growing and deliquesced when the partially dewetted surface was introduced back into vacuum for several days (Supplementary Note 11, Supplementary Fig. 20). Interestingly, the amount of water to which the layers must be exposed in order to destabilize them, is strongly dependent on the halogen ligand of the deposited anions. For example, little change in the morphology of $[B_{12}I_{12}]^{2-}$-based layers was observed over an extended period of several weeks under ambient conditions (see Supplementary Fig. 21) indicating increased layer stability with increase in the halogen size in $[B_{12}X_{12}]^{2-}$. Dewetting of the stable $[B_{12}Br_{12}]^{2-}$ and $[B_{12}I_{12}]^{2-}$-based layers was induced by exposure to a continuous flux of wet gas (Fig. 4, Supplementary Note 12, Supplementary Fig. 22). We found that roughly one order of magnitude more water (longer exposure) was necessary to destabilize the $[B_{12}I_{12}]^{2-}$-based layers than the $[B_{12}Br_{12}]^{2-}$-based layers under the same conditions. The $[B_{12}I_{12}]^{2-}$-based layers developed through several stages of surface morphology with distinctly different optical properties before droplets were finally formed (Fig. 4). It follows that the observed trend of increasing layer stability with increasing size of the halogen atoms is also present for X=Br, I. Water intake destabilizes the layers, but this process has different timescale and effects depending on the deposited anion.

**Influence of surface-layer interaction.** The stability of liquid-like layers is determined by interface potentials[54] and, therefore, the nature of the surface may influence the mechanism and time scale of self-organization of the macroscopic layers. The presence of an interface dipole on FSAMs oriented with the negative pole at the vacuum interface[56,57] led us to suggest that the anion–FSAM interaction potential may be dependent on the internal charge distribution of the anions. Calculated charge distributions of $[B_{12}X_{12}]^{2-}$ (Fig. 3) show that $[B_{12}F_{12}]^{2-}$ carries a large negative charge on the halogen shell of the cluster. Repulsive interaction of the negative cluster shell with the negative interfacial dipole of the

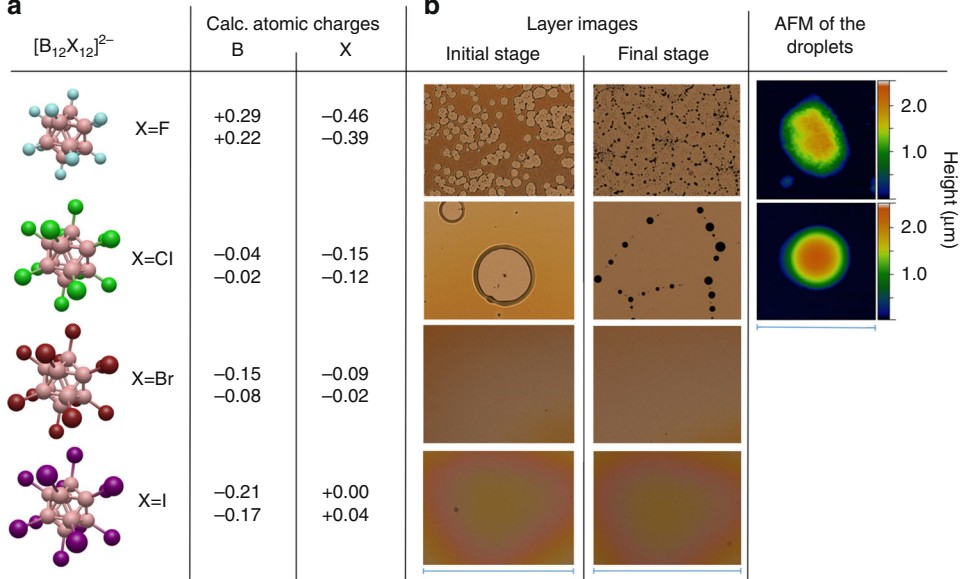

**Fig. 3** Comparison of atomic charges and self-organization of the anion-based layers under ambient conditions. **a** Calculated atomic charges (highest and lowest values)[74] of boron and halogen atoms obtained using different population analysis methods (Natural Population Analysis (NPA) and CHarges from ELectrostatic Potentials using a Grid-based method (CHELPG)) and **b** bright field optical microscopy images (scale bars=540 μm) showing layer morphology after soft landing of the family of $[B_{12}X_{12}]^{2-}$ ions on FSAM. For X=Br, I, the layers were stable and did not change over a timeframe of days, while dewetting was observed for X=F, Cl. The initial stage of hole formation and final stage (after droplet formation) are shown. AFM images (scale bar=15 μm) of the droplets formed in the final stage showing a liquid-like droplet for X=Cl and a droplet with a solid-like grainy morphology for X=F

FSAM may destabilize the layer. The $[B_{12}X_{12}]^{2-}$–FSAM interaction becomes more attractive with decreasing electronegativity of the halogen which reduces the negative charge localized on the shell of the clusters. In addition, a neutral or even slightly positive charge on the polarizable iodine atoms of $[B_{12}I_{12}]^{2-}$ helps rationalize the observed higher stability of the corresponding layers on FSAM. In order to explore the influence of the anion-SAM interaction on the stability of the layers in more detail we conducted equivalent experiments with $[B_{12}X_{12}]^{2-}$ layers on HSAM (undecanethiol, see Supplementary Note 13). In comparison with FSAM, HSAM has a much smaller interface dipole that exhibits a reverse orientation[56], which should reverse the trend in anion-SAM affinity. The final dewetting patterns on HSAMs are distinctly different from those on FSAMs (see Supplementary Figs. 23 and 24) indicating the influence of the surface on the process. However, the dewetting kinetics on FSAM and HSAM show the same trend for the series of $[B_{12}X_{12}]^{2-}$ anions. Similar to FSAM, layers prepared by soft landing of $[B_{12}F_{12}]^{2-}$ onto HSAMs undergo dewetting in seconds under ambient conditions while $[B_{12}I_{12}]^{2-}$ layers are much more stable. Therefore, the dewetting kinetics under ambient conditions are mainly influenced by the anion, which may be responsible for differences in water uptake. The final morphology is not only anion dependent but also strongly influenced by the surface.

**Influence of accumulated molecules from the gas phase**. The differences between the layers described herein were reproduced over a time frame of 1.5 years during which the relative abundance of the identified adventitious hydrocarbons (phthalates) in the deposition chamber changed (see Supplementary Note 14). The distinct differences in the macroscopic layer properties observed for different $[B_{12}X_{12}]^{2-}$ were highly reproducible, and very similar bulk properties of the layers were consistently observed in our experiments independent of the exact composition of the phthalate mixture (Supplementary Fig. 25). By introducing condensed diisodecyl phthalate (Supplementary

Note 15) we were able to substitute the mixture of adventitious phthalates in the layer (Fig. 2b) almost completely with this well-defined molecule (Supplementary Figs. 26, 27).

To change the nature of the co-adsorbed molecules substantially, we also deposited $[B_{12}Cl_{12}]^{2-}$ in the presence of glycine, a molecule with substantial zwitterionic character (≈14 D) in the condensed phase and sufficient vapor pressure to enable co-deposition with the anions[58]. Experimental details are presented in Supplementary Note 16. We conducted two experiments on two different ion soft landing instruments differing in the background pressure in the deposition region and observed a partial (Fig. 5a, see Supplementary Fig. 28) and almost complete (Fig. 5b) substitution of the co-adsorbed adventitious hydrocarbons with glycine. In comparison with $[B_{12}Cl_{12}]^{2-}$ layers containing adventitious hydrocarbons that typically dewet completely in <2 days, much slower dewetting was observed on the surface containing a mixture of the adventitious hydrocarbons and glycine. Remarkably, even after 1 month, dewetting was still incomplete (Fig. 5a). Full substitution of the co-adsorbed adventitious hydrocarbons with glycine eliminated all liquid-like behavior of the deposited layers (Fig. 5b). This solid-like surface showed a grainy morphology. Our results, therefore, provide strong evidence that layer properties can be sensitively tuned by controlling the composition of the background during ion deposition.

**Controlled formation of micro- and nanostructures**. The reported molecular-level control over self-organization of the anion-based layers may be combined with additional processing tools to obtain a final state of the layer with tailored properties. For example, due to their water-sensitive nature, the self-organization of the layers stops under dry conditions (storage under 99.99% $N_2$). The borders of the holes stop expanding and their shape may be preserved. Subsequent exposure of the partially developed layer to water vapor results in dewetting outside of the borders (Fig. 6a). Furthermore, we found that a

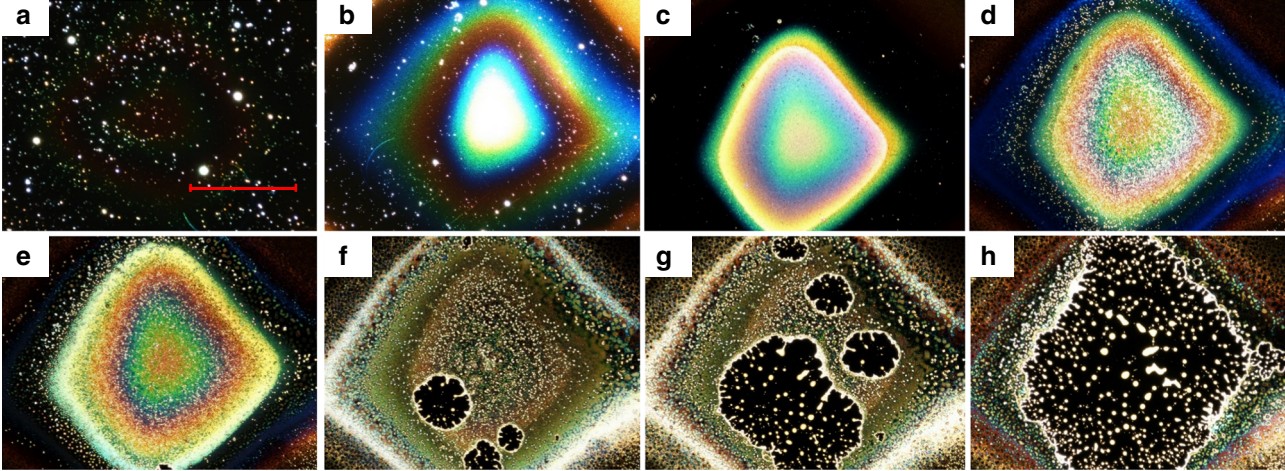

**Fig. 4** Change in the optical properties of the $[B_{12}I_{12}]^{2-}$-based layer during water induced dewetting. Dark field optical microscopy images (scale bar=600 μm) showing the change in the $[B_{12}I_{12}]^{2-}$ layer on FSAM induced by continuous intake of water. **a** before the experiment a smooth layer is not visible in dark field, **b–e** continuous change of the surface morphology during water intake, **f–h** layer ruptures and holes grow. Droplets are formed on the surface

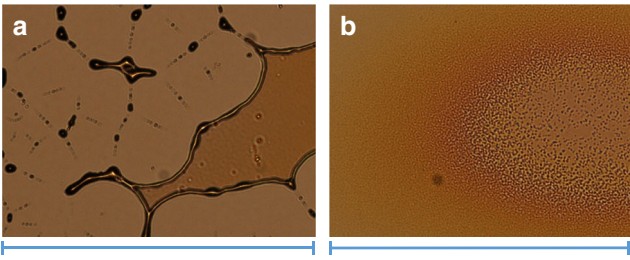

**Fig. 5** Influence of the accumulated neutral molecule. Optical microscopy images (scale bars 540 μm) of surfaces generated by ion soft landing of $[B_{12}Cl_{12}]^{2-}$ with **a** partial and **b** complete substitution of the adventitious hydrocarbons with sublimed glycine. **a** Partial glycine substitution led to slower and incomplete dewetting (compare with Fig. 3b), **b** full substitution with glycine resulted in a stable grainy layer that showed no change under ambient conditions

focused electron beam may be used for direct writing of structures in the layers[59]. High energy electron beams and resulting low energy secondary electrons may induce numerous reactions[60], such as polymerization of the organic components of the layer. In this experiment, we irradiated a part of a layer containing a growing hole with an electron beam arranged spatially into a grid pattern (Fig. 6b). Dewetting of the layer after the electron beam irradiation removed the material away from the grid pattern leaving behind free standing tips at the positions of electron irradiation. Higher vertically protruding tips were produced at the position of the hole borders, and smaller tips at the position of the layer indicating the effect of the layer thickness on the size of the nanostructures produced by the electron beam. The tips were coated with the material of the layer (see Fig. 6d and Supplementary Note 17, Supplementary Fig. 29).

## Discussion

We consider the negative electrostatic charge that accumulates on the surface during soft landing of ions of one polarity in the vacuum system to attract polar molecules with low vapor pressure to form a stabilizing medium for the deposited anions. Although water is an abundant component of our instrument background, we did not detect any measurable amounts of water in the generated layer under vacuum. We propose that the relatively low binding energy of water molecules to dodecaborates precludes

their retention in the layer prepared at room temperature. Furthermore, the recently reported affinity of dodecaborates towards hydrophobic binding pockets[61] may be reponsible for the strong interactions with phthalates in the layer (see a comparative computational investigation for binding of phthalates and water molecules to $[B_{12}Cl_{12}]^{2-}$ in Supplementary Note 18, Supplementary Fig. 30-32). However, we observe that these hygroscopic layers accumulate moisture under ambient conditions, which initiates the dewetting process. The phenomenon of dewetting arises from the interplay of unfavorable surface interactions and attractive intermolecular forces[45,62]. The intake of water into the layer results in a shift in surface energy, which contributes to a higher contact angle of the material on the hydrophobic SAM surfaces. We note that phthalates – a major component of the layer - have high contact angles themselves on FSAM (Supplementary Note 19, Supplementary Fig. 33). However, a smooth layer containing phthalates and $[B_{12}X_{12}]^{2-}$ anions is initially formed in our soft landing experiments. This indicates that the deposited anions are responsible for a favorable interaction of the phthalate-containing layers with the surface, which is eliminated or overcompensated by the intake of water. The capacitor model (see Supplementary Fig. 1) provides an explanation for such an attractive force very similar to the model used to explain the technical process called electrowetting[63], which reduces contact angles of liquids on charged surfaces. This attractive force becomes less pronounced if charge balancing within the layer proceeds during water intake. This may rationalize the slow and controlled dewetting processes observed. Although the cluster anions examined in this study only differ in their halogen, the time scale and mechanism of the self-organization process was found to be strongly dependent on the deposited anion. The amount of water exposure necessary to destabilize these layers follows the trend F<Cl<Br<I. This trend may be attributed to differences in the anion–water interaction that determines the water uptake efficiency by the layer. Ion dipole interactions with water are much stronger for small and hard ions (Pearson concept)[64] than for large and soft ones. Recent studies revealed an exceptionally strong interaction of $[B_{12}F_{12}]^{2-}$ with polar neutral molecules in the gas phase[65,66]. This is in agreement with the strong attraction of $[B_{12}F_{12}]^{2-}$-based layers towards water observed here. In contrast, the different charge distribution of $[B_{12}I_{12}]^{2-}$ may led to weaker binding of water (see Supplementary Note 20, Supplementary Fig. 34, Supplementary Tables 3 and 4 for details of computational investigations).

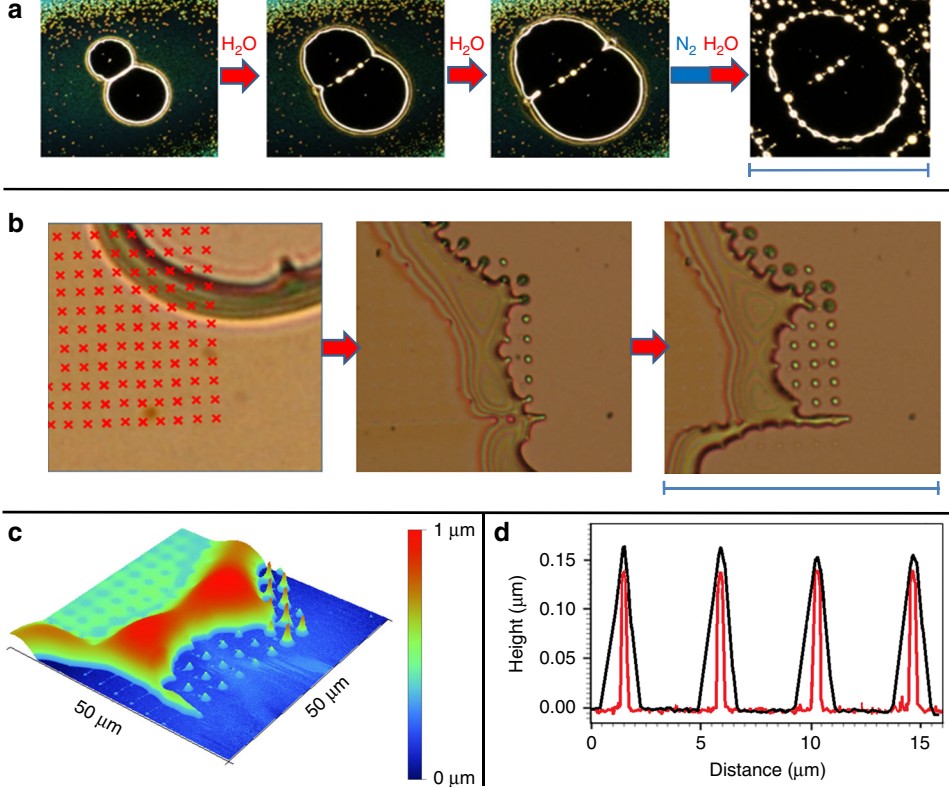

**Fig. 6** Controlled structure formation. **a** Optical microscopy images (scale bar 430 μm) of a controlled dewetting experiment with a $[B_{12}Br_{12}]^{2-}$-based layer. Exposure to wet gas induced formation of two neighboring holes that expanded during further exposure to water vapor. The wet gas flow was interrupted to acquire images a1–a3, which immediately stopped growing. Growth of the holes continued when the surface was re-exposed to water vapor. After picture a3 was taken, the surface was stored for one night under nitrogen and again exposed to the wet gas flow. **b** Optical microscopy images (scale bar=68 μm). Part of the layer containing a growing hole was exposed to a focused electron beam (10 keV, 1.7 nC per irradiated spot) in the vacuum environment of a SEM. The red raster markers indicate the positions of e-beam irradiation. The development of the irradiated surface during further dewetting is shown in the following optical images. **c** AFM image showing the 3D morphology of the last optical image in **b**. **d** AFM line profile over four neighboring tips (short free standing tips in **c** after the material front has passed (black) and after the surface was sonicated in methanol (red, see details in Supplementary Fig. 29), which dissolves non-irradiated areas of the deposit. The comparison shows that the electron beam induced shape is coated by the moving front of deposited liquid-like material

During the formation of a classical condensed phase ionic material, ions of opposite charge usually arrange in an alternating order to minimize repulsion between like-charges. Therefore, strong differences in the interaction of the individual ions with neutral molecules (for example water) are mostly shielded by counterions. As expected, $[B_{12}X_{12}]^{2-}$ salts dissolved and drop-casted on FSAM and HSAM from methanol solution formed small colorless crystals on the surface following solvent evaporation and did not show any obvious reaction resulting from exposure to moisture (see Supplementary Note 21, Supplementary Fig. 35). We conclude that the macroscopic properties of the layers generated by accumulation of mass selected $[B_{12}X_{12}]^{2-}$ gas phase ions are strongly affected by the properties of the unshielded anions that are typically relevant only for isolated gaseous ions. High melting temperatures of dodecaborate ionic compounds are attributed to strong electrostatic forces between the doubly charged anions and countercations, while the phase of the dodecaborate-based layers formed in these experiments seems to be mainly determined by the layer component that accumulates from the gas phase (phthalates: liquid, glycine: solid). The combination of direct writing (Fig. 6) and self-organization in these layers open new vistas in the design of interfaces for advanced technological applications. Future studies will focus on obtaining a detailed understanding of all charge balancing processes in the formed layers both during ion deposition and ambient dewetting process, which will help to exploit the full potential of these materials.

To conclude, we have provided evidence for the controlled generation of macroscopic quantities of a previously unreported type of material on surfaces through deposition of stable mass-selected anions and co-adsorption of molecules from the gas phase. For the first time, the classical molecular structure analysis method NMR was used to analyze mass-selected ions collected on a substrate at the end of a mass spectrometer. The properties of the discovered anion-based layers differ from classical salts, in which strong interactions with cations shield intrinsic anion properties and usually result in the formation of highly stable solids. Instead, deposited anions attract low volatility polar molecules during deposition in vacuum, which stabilizes the layer. Under ambient conditions, the hygroscopic layers undergo a self-organization process that is sensitive to the charge distribution in the deposited anions. Both liquid-like and solid-like layers may be produced from deposited anions by selecting the co-deposited molecules. Furthermore, surface properties, coverage and water uptake by the layers have a pronounced effect on stability, kinetics of dewetting, and microscopic structure formation under ambient conditions. These findings open up interesting opportunities to use the extensively investigated properties of ions and clusters in the gas phase for the generation of functional layers with controlled morphology.

## Methods

**Mass spectrometry analysis**. MS analysis was performed using a (LTQ-Orbitrap XL, Thermo Scientific, Bremen, Germany). The resolving power was $m/\Delta m$ 60,000 at $m/z$ 400. Methanol was used as the working solvent. Fragmentation reactions were initiated by collision induced dissociation in a linear ion trap. A nano-DESI source (custom designed)[52] was attached to the MS inlet. Two fused silica capillaries ($50 \times 150 \ \mu m^2$, ID×OD, Polymicro Technologies) were independently positioned using a high-precision micromanipulator (XYZ 500 MIM, Quarter Research, Bend, Oregon). Two Dino-Lite digital microscopes (AnMo Electronics Corporation, Sanchong, New Taipei, Taiwan) were used to monitor the mass spectrometer inlet and the relative position of the capillaries with respect to each other. The custom-designed nano-DESI sample holder was attached to a motorized XYZ stage (Zaber Technologies, Vancouver, Canada) and the sample was fixed in place. A syringe pump (Legato 180, KD Scientific) delivered methanol through the primary capillary to the sample. The flow rate was $0.5 \ \mu l \ min^{-1}$. A negative voltage of 3.0 kV was applied to the metal part of the syringe to produce ions via electrospray ionization from the secondary capillary positioned at the mass spectrometer inlet. The heated capillary was maintained at 250 °C.

**Optical microscopy**. Optical microscopy was performed using a Nikon Eclipse LV150 optical microscope in bright field and dark field modes. Images were recorded with the NIS Elements Imaging Software 3.22 using the default settings.

**Scanning electron microscopy**. Scanning electron microscopy was performed with a Quanta 3D model, (ThermoFisher Scientific, Inc.) in secondary electron mode with an EDAX energy dispersive X-ray (EDX) spectrometer and a Si(Li) detector with a $10 \ mm^2$ active area.

**XPS**. For the XPS measurements, a Physical Electronics Quantera Scanning X-ray Microprobe equipped with a focused monochromatic Al Kα X-ray (1486.7 eV) source for excitation was used. The instrument contains a spherical section analyzer and a 32 element multichannel detection system. The angle between the X-ray beam (incident normal to the sample) and the photoelectron detector is 45°. A pass-energy of 69.0 eV with a step size of 0.125 eV were used for collecting the high energy resolution spectra. These conditions produced a FWHM of 0.93 eV ± 0.05 eV for the Ag $3d_{5/2}$ line. The Cu $2p_{3/2}$ signal at 932.62 ± 0.05 eV and Au $4f_{7/2}$ at 83.96 ± 0.05 eV were used for calibrating the binding energy scale.

**AFM**. Atomic force microscope (AFM) imaging was performed with a Veeco Dimension Icon instrument. AFM data in Fig. 3, and in the Supplementary Figures 3, 4, 5, 18, 19 and 29b were collected in tapping mode using a Bruker RTESPA-300 cantilever tip (40 N/m spring constant, 8 nm tip radius). AFM data in Fig. 6 and Supplementary Figure 29a were collected in contact mode at 9.5 nN constant force using a Bruker MLCT cantilever tip (0.04 N/m spring constant, 20 nm tip radius).

**IRRAS**. A Bruker Vertex 70 FTIR spectrometer (Bruker Optics, Billerica, MA) equipped with a liquid nitrogen cooled mercury–cadmium–telluride (MCT) detector was used for the IRRAS experiments (grazing incidence) on a SAM sample under vacuum at $6 \times 10^{-5}$ Torr. The detected infrared light is deflected by a flat gold-coated mirror. In front of the detector, the beam is focused using a 90° gold-coated parabolic mirror with diameter of 50 mm and focal length of 400 mm. A mid-infrared (MIR) KRS-5 wire grid polarizer is located in the beam path in front of the vacuum system. The beam enters the vacuum system through a wedged ZnSe vacuum viewport (Laser Optex Inc., Beijing, China). The surface is located in the focal point of the parabolic mirror. The light reflected from the surface exits the vacuum system through a second ZnSe viewport. A second parabolic mirror focused the light onto the MCT detector. The IR beam path was purged with dry nitrogen gas.

**NMR**. 1-D $^{11}$B and $^1$H NMR spectra were measured at 25 °C on a Varian VNMRS spectrometer operating at a field strength of 17.6 T ($^1$H $\nu_0 = 748.4$ MHz, $^{11}$B $\nu_0 = 240.1$ MHz) with a Varian 5 mm direct, broadband tunable, PFG probe. The $^{11}$B experiments were acquired using a CPMGT2 (VNMRJ) pulse sequence consisting of an initial 90° pulse followed by two 180° spin-echoes (RD-90°-τ-180°-2τ-180°-τ-acquire) with a relaxation delay (RD) of 500 ms and τ-delay of 50 μs. The CPMGT2 pulse sequence was employed to help attenuate the high $^{11}$B background signal emanating from both the probe (borosilicate glass probe insert) and borosilicate glass NMR tubes. The spectral width was 100 kHz and 40,000 complex points were acquired in 200 ms and zero-filled to 131,072 prior to Fourier transform. A total of 16384 transients (3 h 11 minute total experiment time) were co-added for each $^{11}$B analyte spectrum. A left-shift of the FIDs by 60 points (300μs) was performed as an additional means of attenuating the high background $^{11}$B signal from both the probe (borosilicate glass probe insert) and borosilicate glass NMR tubes and improve the baseline. Other post-acquisition processing included backward linear prediction using 32 coefficients and 256 basis points, exponential multiplication (22 Hz line broadening), and baseline correction. The 90° pulse width was calibrated using a sample of $[HN(CH_3)_3]_2[B_{12}I_{12}]$ dissolved in fully deuterated

methanol ($CD_3OD$). An external 189 mM boric acid sample in $D_2O$, having a measured pD of 7.8 (pD = pH + 0.4) was used to reference the spectra and its chemical shift was estimated to be +18.0 ppm[67]. The $^1$H NMR spectrum was measured using a standard 1D pulse sequence. The spectral width was 12 kHz (144230 complex points acquired in 6 s and zero-filled to 524288 points), and 256 transients were acquired with a relaxation delay of 10 s. The $^1$H spectrum was referenced externally to the signal of tetramethylsilane (TMS) in $CD_3OD$. The $^1$H NMR spectrum was measured using a standard 1D pulse sequence. The spectral width was 12 kHz (144230 complex points acquired in 6 s and zero-filled to 524288 points), and 256 transients were acquired with a relaxation delay of 10 s and post-acquisition processing included exponential multiplication (1 Hz line broadening). The $^1$H spectrum was referenced externally to the signal of tetramethylsilane (TMS) in $CD_3OD$.

**Molecular modeling**. Computational investigations were performed using the NWChem[68] software. All structures were optimized with DFT (PBE0[69]/def-tzvppd)[70] and frequency analyzes were performed.

**MicroXRD**. A Rigaku D/Max Rapid II instrument with a 2D image plate detector was used for MicroXRD investigations. A MicroMax 007HF generator fitted with a rotating Cr anode (Lambda = 2.2897 Å) generates X-rays which are focused on the specimen through a 300 um diameter collimator. Integration of the diffractions rings captured by the detector was performed using 2DP, Rigaku 2D Data Processing Software (Ver. 1.0, Rigaku, 2007) to obtain the patterns in intensity vs. 2-Theta form. For analysis of data JADE v9.5.1 (Materials Data Inc.), and PDF4 +database from ICDD were used.

**Vacuum conditions**. A detailed description of the soft landing instrument can be found in ref [42]. Except for the glycine experiments and substitution with a defined phthalate, depositions were performed at a pressure of $5.5$–$5.8 \times 10^{-5}$ Torr. The base pressure of the instrument, with the inlet closed is $1.0 \times 10^{-5}$ Torr. Pressures were measured with an ion gauge located in the deposition chamber. Vacuum chambers housing the dual ion funnel interface and collisional quadrupole are evacuated by rotary vane mechanical pumps. The deposition region is evacuated by a turbo pump (Agilent TV301) backed by a rotary vane mechanical pump. All of the mechanical pumps use "Permavis 10" pump oil purchased from the Kurt J. Lesker Company (Jefferson Hills, PA). O-Rings used as seals in the instrument were obtained from Pfeiffer Vacuum and McMaster-Carr (Elmhurst, IL).

**Synthesis**. Alkali metal salts of the halogenated closo-dodecaborate were synthesized by reactions of the corresponding salts of the parent $[B_{12}H_{12}]^{2-}$ anion with elemental halogens according to known procedures. Fluorination with $F_2/N_2$ in $CH_3CN$ gave $[B_{12}F_{12}]^{2-}$[71], Chlorination in boiling water yielded $[B_{12}Cl_{12}]^{2-}$[50], and bromination with bromine in aqueous methanol produced $[B_{12}Br_{12}]^{2-}$[72]. The $[B_{12}I_{12}]^{2-}$ dodecaborate was obtained by a microwave-assisted reaction with iodine in acetic acid[73]. Subsequent metathesis reactions with ammonium halides gave the materials used in the ion soft-landing experiments.

**Formation of self-assembled monolayer surfaces on gold**. $1H,1H,2H,2H$-Perfluorodecanethiol (FSAM), 1-undecanethiol (HSAM) and the solvent, methanol, were purchased from Sigma-Aldrich. The gold substrates were purchased from Platypus Technologies (Madison, WI) and have the following specifications: $10 \times 10 \ mm^2$, 525 μm thick Si, 50 Å Ti adhesion layer, 1000 Å Au layer. The gold substrates were washed in methanol in an ultrasonic bath for several minutes, dried under nitrogen, further purified with a Boekel (Boekel Scientific, Feasterville, PA) ultraviolet cleaner, and then immersed for 12 h in glass scintillation vials containing a 1 mM methanol solution of the corresponding thiol for formation of the SAM. Subsequently, the surfaces were ultrasonically washed for a half minute in methanol. Then, the surfaces were rinsed with methanol, dried with a flow of nitrogen ($N_2$), and mounted in the sample holder of the soft-landing machine.

**Data availability**. Data availability on request from the authors.

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

## Acknowledgements

J.W. acknowledges a Feodor Lynen Fellowship from the Alexander von Humboldt Foundation, support from the Pacific Northwest National Laboratory (PNNL) through the Alternate Sponsored Fellow Program and he thanks Vladimir A. Azov for very helpful discussions. G.E.J. and J.L. acknowledge support from the U.S. Department of Energy (DOE), Office of Science, Office of Basic Energy Sciences, Division of Chemical Sciences, Geosciences and Biosciences. A portion of the research was performed using EMSL, a DOE Office of Science User Facility sponsored by the Office of Biological and Environmental Research. PNNL is a multiprogram national laboratory operated for DOE by Battelle. E.A. used resources provided by PNNL Institutional Computing (PIC). We thank Marshall Ligare and Venky Prabhakaran for technical support, Tamas Varga for the micro-XRD measurements, Ruichuan Yin for help with nano-DESI measurements and Ziyan Warneke for assistance with preparing graphical material.

## Author contributions

J.W.: Conceived and developed the principle idea to use dodecaborates in ion soft landing, performed the experiments, analyzed the data, performed molecular modeling and wrote the manuscript. M.E.M.: Performed AFM measurements and prepared corresponding images. S.L.R.: Performed AFM measurements and prepared corresponding images. S.C.: Performed SEM-imaging experiments and electron beam patterning. M.H. E.: Performed XPS measurements and corresponding data analysis. E.A.: Performed molecular modeling. R.P.Y.: Performed NMR experiments, data analysis and prepared corresponding graphical material. N.M.W.: Performed NMR experiments and data processing. C.J.: Synthesized samples and provided expertize in boron chemistry. G.E.J.: Designed and performed the experiments, analyzed data, and co-wrote the manuscript. J. L.: Supervised the project, designed the experiments, performed nano-DESI measurements, analyzed data and co-wrote the manuscript.
