## [Peer Review File · Nature Communications]

Reviewers' comments:

Reviewer #1 (Remarks to the Author):

Laskin et al report an interesting phenomenon in which self-organization of mass-selected anions can be triggered by the presence of co-adsorbed neutral molecules. Both liquid-like and solid-like dewetting processes were observed by controlling the identity of the co-adsorbing neutral molecule. The uniqueness of this study is not only reflected in the discovery of this new phenomenon and the correct interpretation it, but also the fact it is now possible to soft-land large quantities of highly purified gas-phase ions under vacuum to an extent where an orthogonal, and reproducible but less sensitive, NMR can be used for material characterization. This is exciting for material synthesis, surface modification and analytical instrumentation; one can therefore expect high citation from this work. The high quality write-up could be published as is. However, the current use of the notation "SI" in the text to refer to supplemental figures make it hard to follow. It will be better to use supplemental figure numbers instead. Is glycine actually zwitterionic in the gas-phase, in the absence of solvent molecules?

Reviewer #2 (Remarks to the Author):

In the manuscript "Self-organizing layers from complex molecular anions" by Warneke et al. the investigation of the behavior of anion layers fabricated by ion deposition in vacuum is reported.

The manuscripts main points are the handling of isolated molecular ions in the gas phase to a point that can generate mesoscopic structures on surfaces. This is impressively demonstrated by the deposition of up to $1e15$ ions. However, some more details of the process would be useful, since in the following the behavior of the films are discussed and for that quantities like deposition time and thickness can matter a great deal.

The thin films, due to the special way they were prepared, are certainly novel. The discussion of the origin and evolution of their morphology is described in the context of the properties of the deposited ions, specifically upon exchanging the halogen species in the molecule. With the title and scope of the manuscript suggesting that the self-organization is the focus of the work, very little data on the internal structure of the layers is presented but only light microscopy, which shows a general view of the morphology and dewetting. While its behavior can be correlated with the deposited ion species, the composition of the film remains unclear. Especially the experiments on

co-incorporation of organic species in vacuum and water in ambient/wet-air environments or also glycine are often not completely and comprehensively described, lack a quantitative analysis, and a molecular picture/model of the structure of the films is not established.

In conclusion, the manuscript presents a very interesting case of material fabrication that could push the boundaries of what is possible at present. In particular the processing of a large amount of ions like presented here has the potential to generate new and interesting properties. Unfortunately, the characterization of the obtained structures remains vague. At present, I can therefore not recommend it for publication, but it should certainly be reconsidered if more experimental evidence towards the structure and composition of the fabricated layers is presented.

DETAILED COMMENTS

-1-

(P1L40) The statement of a first observation is hardly helpful, in particular in light of several observations of the effect of charged constituents on the morphology of films.[1,2] Interestingly both works show a phenomenon on the molecular scale that could be considered dewetting.

(P2L53) Similarly, the preparation of multilayered thin films is reported in the literature [3,4][weitzel] also using intense deposition sources [5].

While the results referenced here may not have been achieved under the exact same circumstances as the presented result, they deal with the same phenomenon and should be put into context.

-2-

The composition of the fabricated layers is of crucial importance to this study. Indeed the study delivers results to this end, but the phrasing sometimes seems to lack clarity or context. For instance P2L100 mentions "layers of complex ions", while later in the manuscript the coadsorption of neutral molecules is investigated and discussed. This referee recommends finding a clear terminology to use throughout the manuscript.

-3-

At many points the "intrinsic property of the deposited anions" is mentioned (P1L19/20, P2L41, P2L74, P5L138, P6L194, P10L299). What is exactly meant by this other than that they are charged? I

can imagine that this includes properties like reactivity and dielectric properties, but it is not well-defined in the manuscript.

-4-

The use of nano DESI for the chemical characterization of the layers will certainly result in incomplete results. In particular, the main component of the background pressure in the vacuum in the $1\text{e-}6$ mbar range as used here is mainly water. (Most likely the pressure of $1\text{e-}5$ mbar given for the glycine experiment is water too, which is driven out of the powder of the evaporator – unless it was degassed beforehand well below the sublimation temperature on glycine.) While certainly also hydrocarbons are introduced into the layers during growth, the main component will be water, which is not detected in DESI.

-5-

The description of the glycine experiment is ambiguous. Is the the 70C heating or 200C heating intended for vaporizing or purifying the molecules? At which pressure is this done? Is the vapor introduced in the chamber or the powder?

-6-

Directly related to the point of the chemical composition of the films is the question after the volume of the material observed in the different stages. While the presence of $1\text{e}15$ ions ($z=-2$) on a spot of 3mm diameter suggests a film thickness of approx. 10 nanometers, the optical images obtained show interference fringes, corresponding to significant swelling reaching the dimensions of the wavelength. So there is a significant uptake of material, which is however not quantified.

Also the deposition of this amount of material requires around 100 nAh of charge, which will take a significant amount of time. For a very hygroscopic material, given such a long time in a vacuum chamber of partial pressure of $1\text{e-}6$ mbar in water, the material incorporation can be quite significant there already as the monolayer time at this pressure is approx. 1 second assuming sticking coefficient 1. So could the liquid layers of B12X12 be in fact solutions (P7L233).

-7-

The final part showing the electron beam modification of the layers feels very detached. In particular, with the composition of the layers largely unclear, the chemical reactions imposed by a 10keV focused electron beam are manifold and thus it is not surprising that something happens. Clearly, the further handling of a film produced by the presented method is of great relevance, however, it should be put into the context of the other experiments.

REFERENCES

[1] Della Pia, A.; Riello, M.; Floris, A.; Stassen, D.; Jones, T. S.; Bonifazi, D.; De Vita, A. & Costantini, G.: Anomalous coarsening driven by reversible charge transfer at metal--organic interfaces. *ACS Nano*, ACS Publications, 2014, 8, 12356-12364

[2] Fernandez-Torrente, I.; Monturet, S.; Franke, K. J.; Fraxedas, J.; Lorente, N. & Pascual, J. I.: Long-Range Repulsive Interaction between Molecules on a Metal Surface Induced by Charge Transfer. *Phys. Rev. Lett.*, 2007, 99, 176103

[3] Rauschenbach, S.; Rinke, G.; Malinowski, N.; Weitz, R. T.; Dinnebier, R.; Thontasen, N.; Deng, Z.; Lutz, T.; de Almeida Rollo, P. M.; Costantini, G.; Harnau, L. & Kern, K.: Crystalline Inverted Membranes Grown on Surfaces by Electro spray Ion Beam Deposition in Vacuum. *Adv. Mater.*, 2012, 24, 2761-2767

[4] Menezes, P.; Martin, J.; Schäfer, M.; Staesche, H.; Roling, B. & Weitzel, K.-M. Bombardment induced ion transport—Part II. Experimental potassium ion conductivities in borosilicate glass. *Physical Chemistry Chemical Physics*, 2011, 13, 20123-20128

[5] Pauly, M.; Sroka, M.; Reiss, J.; Rinke, G.; Albarghash, A.; Vogelgesang, R.; Hahne, H.; Kuster, B.; Sesterhenn, J.; Kern, K. & Rauschenbach, S.

A Hydrodynamically Optimized Nano-electrospray Ionization Source and Vacuum Interface

Analyst, 2014, 139, 1856 - 1867

Reviewer #3 (Remarks to the Author):

The authors report the characterizations of phases, self-organization mechanisms and optical properties for layers of anions on surfaces, formed through deposition of stable mass-selected anions of dianionic halogenated dodecaborates [B₁₂X₁₂]²⁻ (X = F, Cl, Br, I). They bridge between microscopic information and macroscopic observation, with infrared, X-ray and mass spectroscopy and with optical images. In this manuscript, “dewetting” is a key issue to characterize the controlled anion layers, and the findings are discussed in terms of self-organization, charge distributions and co-deposition of neutral molecules. Although the dewetting resulted from molecular-level controllability seems interesting phenomenologically, the scientific understanding is insufficient to deepen the knowledge in physics and chemistry.

As stated in the manuscript, co-adsorbed adventitious hydrocarbons seemingly play an important role of compensation of ionic charge balance to prevent the deposited anions from escaping repulsively against Coulombic repulsions. The convincing characterization of the co-adsorbed adventitious hydrocarbons is indispensable with the microscopic discussion; oil vapor? At this stage, another research group cannot reproduce the experiment.

More Importantly, to reveal the morphology change with co-deposition of neutral molecules and with water uptake under ambient conditions, in situ AFM measurements is inevitable because the authors discuss about the dewetting without any microscopic information on initial morphology. Depending on the coverage and the kind of halogen X for B₁₂X₁₂(²⁻), the initial morphology regarding uniformity might be changed and the image shown in Fig. S1-c) is not guaranteed experimentally.

The reviewer recommends the Editor not to accept the manuscript in Nature Communications. Since the manuscript is well written literally, it should be published in a more specialized journal relevant to applied physics after the revisions indicated below.

[Major points]

- (1) The convincing characterization of the co-adsorbed adventitious hydrocarbons is indispensable.
- (2) On initial morphology, in situ AFM measurements is inevitable to identify uniformity molecularly.

[Minor points]

L24; In abstract, the authors state “rational control of macroscopic properties”, but the meaning of macroscopic properties is not clear.

L95-96; The thickness of anionic layer should be defined, corresponding to 1013, 1014 and 1015 ions. Although it is not easy to define the monolayer (ML), the amount of ions is not a good index for the thickness. Depending the deposition area size, the local thickness will be varied.

L102; In Fig. 1 b), height is changed with time?

L63; Although the chemical composition of F-SAM (1H,1H,2H,2H-perfluorodecanethiol is shown in the supplementary information in SI-14, it is better to show the chemical composition of fluorinated thiols for fluorinated self-assembled monolayer surfaces also in the text for the reader.

L108-109 The authors state that “observed IR signals grew continuously during ion deposition in vacuum”. The reviewer wonders if intensity increases linearly against the deposition time without charge repulsions?

L115-116; The authors state that “We propose that adsorption of polar neutral molecules from the instrument background is driven by the presence of large numbers of ions on the surface”. Is it possible to reproduce the adsorption in another laboratory? The contamination may be native to the author’s experimental setup.

L128; The authors state that “the layers are stabilized by co-adsorption of neutral molecules from the gas phase”. From where the neutral molecules come? Is there any evidence that the neutral molecules deposited survive as neutral species?

L141; “background pressure” What is main contamination in residual gas? How do they control the background pressure?

L245; Glycine vapor pressure at room temperature and at heated temperature should be shown.

How much is the base pressure without glycine? The background pressure of $\sim 10^{-5}$ Torr should be compared to the base pressure without glycine. How much is the heating temperature?

L267-275; The discussion about Figure 6 is unclear. The scientific or engineering discussion on Figure 6 should be added to clarify the focus, rather than a long figure caption of Figure 6.

Answer to the Reviewer's comments (NCOMMS-17-22947-T)

Reviewer #1 (Remarks to the Author):

Laskin et al report an interesting phenomenon in which self-organization of mass-selected anions can be triggered by the presence of co-adsorbed neutral molecules. Both liquid-like and solid-like dewetting processes were observed by controlling the identity of the co-adsorbing neutral molecule. The uniqueness of this study is not only reflected in the discovery of this new phenomenon and the correct interpretation it, but also the fact it is now possible to soft-land large quantities of highly purified gas-phase ions under vacuum to an extend where an orthogonal, and reproducible but less sensitive, NMR can be used for material characterization. This is exciting for material synthesis, surface modification and analytical instrumentation; one can therefore expect high citation from this work. The high quality write-up could be published as is. However, the current use of the notation "SI" in the text to refer to supplemental figures make it hard to follow. It will be better to use supplemental figure numbers instead. Is glycine actually zwitterionic in the gas-phase, in the absence of solvent molecules?

Author reply: We thank the reviewer for the positive evaluation of our manuscript. Glycine is not zwitterionic in the gas phase but is in the condensed phase so that glycine is expected to be in the zwitterionic form in the layers. We have clarified this in the corresponding sentence. The notation "SI" usually does not refer to a single figure, but rather to a section, including additional descriptions. Therefore we now use the notation "section SI" instead.

Reviewer #2 (Remarks to the Author):

In the manuscript “Self-organizing layers from complex molecular anions” by Warneke et al. the investigation of the behavior of anion layers fabricated by ion deposition in vacuum is reported.

The manuscripts main points are the handling of isolated molecular ions in the gas phase to a point that can generate mesoscopic structures on surfaces. This is impressively demonstrated by the deposition of up to $1e15$ ions. However, some more details of the process would be useful, since in the following the behavior of the films are discussed and for that quantities like deposition time and thickness can matter a great deal.

The thin films, due to the special way they were prepared, are certainly novel. The discussion of the origin and evolution of their morphology is described in the context of the properties of the deposited ions, specifically upon exchanging the halogen species in the molecule. With the title and scope of the manuscript suggesting that the self-organization is the focus of the work, very little data on the internal structure of the layers is presented but only light microscopy, which shows a general view of the morphology and dewetting. While its behavior can be correlated with the deposited ion species, the composition of the film remains unclear. Especially the experiments on co-incorporation of organic species in vacuum and water in ambient/wet-air environments or also glycine are often not completely and comprehensively described, lack a quantitative analysis, and a molecular picture/model of the structure of the films is not established.

In conclusion, the manuscript presents a very interesting case of material fabrication that could push the boundaries of what is possible at present. In particular the processing of a large amount of ions like presented here has the potential to generate new and interesting properties. Unfortunately, the characterization of the obtained structures remains vague. At present, I can therefore not recommend it for publication, but it should certainly be reconsidered if more experimental evidence towards the structure and composition of the fabricated layers is presented.

Author reply: We thank Reviewer 2 for pointing out the importance and novelty of our work and for making constructive comments. In the revised version of the manuscript and supporting information, we have addressed all the suggestions made by Referee 2. A detailed point-by-point response is presented below. For further analysis of the layers including their composition and thickness, we have reproduced several deposition experiments, performed additional AFM measurements during dewetting, and conducted additional mass spectrometry-tandem-MS, XPS, IR and micro-XRD investigations and data analysis. The adventitious hydrocarbons are now clearly identified to be predominantly phthalates with different chain length, as it was shown from characteristic fragmentation behavior in MS/MS experiments. In micro-XRD investigations of the droplets formed after dewetting and of the intact layers, no signals corresponding to a periodic structure in the material within the measurable length scale were detected. This observation supports our conclusion that the layers are liquid-like. We expanded section S15 (chemical analysis) significantly and added a section (SI 6) that discusses the thickness of the layers and their morphology prior to dewetting. AFM measurements confirmed that the layers exhibit a very smooth surface prior to dewetting.

The main point in the discussion with Referee 2 is the role of water in the layer (see point-by-point comments and answers). We added data that show that water plays only a negligible role in the *in situ*-formation of the layers, but as expected, contributes to dewetting and volume expansion observed in *ex situ* characterization experiments. The deposition rate did not influence the subsequent development of the layer. This is now stated in the manuscript (page 3 above Figure 1).

DETAILED COMMENTS

-1-

(P1L40) The statement of a first observation is hardly helpful, in particular in light of several observations of the effect of charged constituents on the morphology of films.[1,2] Interestingly both works show a phenomenon on the molecular scale that could be considered dewetting.

(P2L53) Similarly, the preparation of multilayered thin films is reported in the literature [3,4][weitzel] also using intense deposition sources [5].

While the results references here may not have been achieved under the exact same circumstances as the presented result, they deal with the same phenomenon and should be put into context.

Author reply: The references are now all cited in the introduction. The authors did not intend to claim that pattern formation based on charged species or multilayer deposition has not been previously reported. However, to the best of our knowledge, this is the first time that a liquid-like bulk phase layer was generated by deposition of such a large number of mass-selected molecular anions. Also, the macroscopic structure formation was identified to be dewetting, which was not reported in previous soft-landing studies but is relevant to many applications. The statement has been rephrased to clarify this point (break between page 1 and 2).

-2-

The composition of the fabricated layers is of crucial importance to this study. Indeed the study delivers results to this end, but the phrasing sometimes seems to lack clarity or context. For instance P2L100 mentions “layers of complex ions”, while later in the manuscript coadsorption of neutral molecules is investigated and discussed. This referee recommends finding a clear terminology to use throughout the manuscript.

Author reply: We have performed additional experiments and data analysis to improve the understanding of the layer composition (see, for example, detailed answers to point -4-). The sentence (P2L100) has been rephrased and we use only “layer” or “anion based layers” consistently throughout the manuscript now.

-3-

At many points the “intrinsic property of the deposited anions” is mentioned (P1L19/20, P2L41, P2L74, P5L138, P6L194, P10L299). What is exactly meant by this other than that they are charged? I can imagine that this includes properties like reactivity and dielectric properties, but it is not well-defined in the manuscript.

Author reply: The charge distribution within the individual molecular ion is the key intrinsic property that determines the strikingly different behavior of the layers prepared with different $[B_{12}X_{12}]^{2-}$ anions, because it influences interactions with neutrals like water (see theoretical investigations in section SI10) or the surface, see explanations on page 6. We rephrased the corresponding sentence in the abstract and introduction to clarify this point.

-4-

The use of nano DESI for the chemical characterization of the layers will certainly result in incomplete results. In particular, the main component of the background pressure in the vacuum in the 1e-6 mbar range as used here is mainly water. (Most likely the pressure of 1e-5 mbar given for the glycine experiment is water too, which is driven out of the powder of the evaporator – unless it was degassed beforehand well below the sublimation temperature on glycine.) While certainly also hydrocarbons are introduced into the layers during growth, the main component will be water, which is not detected in DESI.

Author reply: The authors understand the concern of Referee 2 and the suggestion that water may be a major component of the *in-situ* formed layers. Additional evidence that the dominant neutral component of the layers prepared under vacuum are hydrocarbons, and not water, is required. During buildup of the layers, we performed *in-situ* IR investigations using pristine FSAM as a background. The intensity of the B-X stretching band increases with the amount of $[B_{12}X_{12}]^{2-}$ ions deposited on the surface. Concomitant with these bands, the hydrocarbon signals also increase in proportion during the deposition. The simultaneous accumulation of water would result in a broad IR band in the range of 3200 – 3600 cm^{-1} (see for example NIST spectra) that would be expected to increase continuously with the ion signal and hydrocarbon signal. However, this band is not observed in our experiments. Additional IR spectra showing this more clearly are now presented in section SI5 “Chemical analysis of layers”.

For additional evidence of layer composition we reanalyzed the XPS data. Although the XPS analysis was performed *ex-situ*, the surface was stored under nitrogen after removal from the soft landing instrument and there was no observable dewetting process in the analyzed area. We show in Figure 2 the elemental composition obtained from the XPS analysis. We can be sure that we measure only carbon from the deposited layer and not from the underlying FSAM, because almost no fluorine content (0.1%) is detected (fluorine, however was clearly detected for lower coverage samples). To be able to measure XPS without the influence from the underlying SAM, we deposited 3×10^{15} ions. This was not stated in the manuscript but is now clarified in the Figure 2 caption. The carbon XPS spectra allow further conclusions based on the peak positions. We performed peak fitting to distinguish the contribution of C-H groups and oxygen bound carbons (C-O, C=O) to the total carbon intensity. This analysis is now available in section SI5 and shows in *argumentum e contrario* that almost all the detected oxygen can be assigned to carbon bound-oxygen so that the amount of water in the layer is negligible before dewetting under ambient conditions starts.

The reviewer points out that a substantial amount of the background pressure in the chamber is water. Solvent molecules from the ESI source are another major source of the background pressure in our soft-landing instrument used in this study. We hypothesized that the dipole moment of the molecule plays an important role in determining its propensity of getting incorporated into the layers through a strong ion-dipole binding to the soft-landed anions. However, identical results were obtained using methanol (dipole moment 1.69 D) and acetonitrile (dipole moment 3.92 D). We observed no signs of accumulation

of either acetonitrile or methanol on the surface in the IR spectra. In addition to the dipole moment, binding of neutral molecules to anions may be enhanced by the presence of multiple functional groups that may interact with the anions. We also note that the surface is maintained at room temperature meaning that both adsorption and desorption of small molecules are quite rapid. We propose that water molecules, while present in the vacuum chamber and may get temporarily adsorbed on the surface, cannot compete with larger less volatile hydrocarbons for being incorporated into the layer. The identity of the adventitious hydrocarbons has been clarified, see section SI 5.

In contrast, under ambient conditions the uptake of water is obvious from the already described behavior of the layers under different environmental conditions (nitrogen atmosphere, wet gas flow). We added XPS investigations on a dewetted layer. Although the carbon ratio has only a limited meaning here because the SAM is exposed in large areas between the droplets, the oxygen to boron ratio, which should be independent of the SAM, showed a significant increase in comparison with the layer analyzed prior to dewetting.

All values are given in atomic %

Smooth protected surface: B 12,6%; O 11,3%

Dewetted surface (droplets): B 2,6%; O 8,7%

From these values that are now included in the supporting information we can make a rough estimate that 20-30 water molecules per deposited ion are present in the layer after dewetting (note that one ion contains 12 boron atoms). Corresponding comments are added on page 4 and page 7 (top). Together with the large amount of detected hydrocarbons, this rationalizes the significant swelling of the material that Reviewer 2 commented on in -6-.

-5-

The description of the glycine experiment is ambiguous. Is the the 70C heating or 200C heating intended for vaporizing or purifying the molecules? At which pressure is this done? Is the vapor introduced in the chamber or the powder?

Author reply: In chapter SI 12, the description has been subdivided into two sections: 1. Partial substitution and 2. Full substitution to make it more clear. The background pressures for both experiments are now provided.

In the case of the partial substitution of hydrocarbons by glycine at $8 \cdot 10^{-5}$ Torr, the powder was introduced into the vacuum chamber and stored on a metal flange directly below the surface. This metal flange was heated. This procedure is described in the supporting information.

We have added more information about the experiment at 200C. Another vacuum chamber with a base pressure of 10^{-8} Torr was used. The glycine reservoir was connected through a leak valve to the main chamber close to the surface. The temperature was slowly increased (over the timeframe of several hours) until a residual gas analyzer could detect glycine in the main chamber, which happened at 200C. At the same time the pressure increased to 10^{-6} Torr.

-6-

Directly related to the point of the chemical composition of the films is the question after the volume of the material observed in the different stages. While the presence of $1e15$ ions ($z=-2$) on a spot of 3mm diameter suggests a film thickness of approx. 10 nanometers, the optical images obtained show interference fringes, corresponding to significant swelling reaching the dimensions of the wavelength. So there is a significant uptake of material, which is however not quantified.

Also the deposition of this amount of material requires around 100 nAh of charge, which will take a significant amount of time. For a very hygroscopic material, given such a long time in a vacuum chamber of partial pressure of $1e-6$ mbar in water, the material incorporation can be quite significant there already as the monolayer time at this pressure is approx. 1 second assuming sticking coefficient 1. So could the liquid layers of B12X12 be in fact solutions (P7L233).

Author reply: Additional information given in -4- addresses part of this question. We provided evidence that water is not a major component in the *in-situ* layers. The amount of hydrocarbons was found to grow in proportion with the amount of deposited ions. In addition, the observed self-organization behavior and droplet size showed only a coverage dependence and no dependence on the timeframe in which this coverage was achieved. At a pressure of 10^{-5} Torr, the amount of collisions of residual gases with the surface is orders of magnitude higher than the ion deposition rate suggesting that the intake of the hydrocarbons is not the determining kinetic factor for the growth of the layer.

A considerably larger volume of material observed in the experiment compared to the estimation from the pure number of ions suggested by Referee 2 can be rationalized by the elemental composition obtained from XPS. Taken into account that every ion contains 12 boron atoms, the number of hydrocarbon units (like CH_2) per ion can be estimated to be roughly 60. Under ambient conditions during the dewetting process and droplet formation, additional water uptake has already taken place as discussed above. We have performed additional AFM investigations on $[B_{12}Cl_{12}]^{2-}$ based layers of different thickness during dewetting that enabled us to estimate film thicknesses. AFM line scans on layers during dewetting that reach from the inside of a hole over the moving material front to the smooth exterior layer are shown in section SI6 of the supporting information that describes layer thickness and morphology. From these investigations we estimate the thickness of the layer in the middle of the deposition spot to be roughly 55 nm and 145 nm after soft-landing of $9 \cdot 10^{14}$ ions and $3 \cdot 10^{15}$ ions, respectively. In the revised version of the manuscript, we refer to these results on page 8 (bottom).

It is, therefore, obvious that the finally formed layers contain a considerably larger amount of co-adsorbed molecules than of deposited ions. Mass selected anions deposited in larger amounts, build up a medium out of available polarizable neutrals with low vapor pressure *in-situ* and accumulate additional water *ex-situ*. We believe that these additional experiments and data analysis fully address the comments made by Reviewer 2. In the revised version of the manuscript, we modified the discussion and conclusions sections to provide deeper insights into the nature of the layer at different stages of development. The striking main point of this manuscript is that the evolution of these layers to the finally dewetted surface is controlled on a molecular level by the charge distribution of the deposited anion.

-7-

The final part showing the electron beam modification of the layers feels very detached. In particular, with the composition of the layers largely unclear, the chemical reactions imposed by a 10keV focused

electron beam are manifold and thus it is not surprising that something happens. Clearly, the further handling of a film produced by the presented method is of great relevance, however, it should be put into the context of the other experiments.

Author reply: The text describing the controlled structure formation using electron beam irradiation has been expanded and clarified. The composition of the layer is now better described. The treatment of the organic material, present in a large amount in these layers, with an electron beam is known to initiate polymerization reactions, which may be responsible for the formation of the tips. However, the combination of this tool for controlled structure formation together with the self organization of the layer that is influenced by the ion properties opens up interesting opportunities that are outlined in the text.

REFERENCES

- [1] Della Pia, A.; Riello, M.; Floris, A.; Stassen, D.; Jones, T. S.; Bonifazi, D.; De Vita, A. & Costantini, G.: *Anomalous coarsening driven by reversible charge transfer at metal--organic interfaces. ACS Nano, ACS Publications, 2014, 8, 12356-12364*
- [2] Fernandez-Torrente, I.; Monturet, S.; Franke, K. J.; Fraxedas, J.; Lorente, N. & Pascual, J. I.: *Long-Range Repulsive Interaction between Molecules on a Metal Surface Induced by Charge Transfer. Phys. Rev. Lett., 2007, 99, 176103*
- [3] Rauschenbach, S.; Rinke, G.; Malinowski, N.; Weitz, R. T.; Dinnebier, R.; Thontasen, N.; Deng, Z.; Lutz, T.; de Almeida Rollo, P. M.; Costantini, G.; Harnau, L. & Kern, K.: *Crystalline Inverted Membranes Grown on Surfaces by Electro spray Ion Beam Deposition in Vacuum. Adv. Mater., 2012, 24, 2761-2767*
- [4] Menezes, P.; Martin, J.; Schäfer, M.; Staesche, H.; Roling, B. & Weitzel, K.-M. *Bombardment induced ion transport—Part II. Experimental potassium ion conductivities in borosilicate glass. Physical Chemistry Chemical Physics, 2011, 13, 20123-20128*
- [5] Pauly, M.; Sroka, M.; Reiss, J.; Rinke, G.; Albarghash, A.; Vogelgesang, R.; Hahne, H.; Kuster, B.; Sesterhenn, J.; Kern, K. & Rauschenbach, S.
A Hydrodynamically Optimized Nano-electrospray Ionization Source and Vacuum Interface Analyst, 2014, 139, 1856 - 1867

Reviewer #3 (Remarks to the Author):

The authors report the characterizations of phases, self-organization mechanisms and optical properties for layers of anions on surfaces, formed through deposition of stable mass-selected anions of dianionic halogenated dodecaborates [B₁₂X₁₂]²⁻ (X = F, Cl, Br, I). They bridge between microscopic information and macroscopic observation, with infrared, X-ray and mass spectroscopy and with optical images. In this manuscript, “dewetting” is a key issue to characterize the controlled anion layers, and the findings are discussed in terms of self-organization, charge distributions and co-deposition of neutral molecules. Although the dewetting resulted from molecular-level controllability seems interesting phenomenologically, the scientific understanding is insufficient to deepen the knowledge in physics and chemistry.

As stated in the manuscript, co-adsorbed adventitious hydrocarbons seemingly play an important role of compensation of ionic charge balance to prevent the deposited anions from escaping repulsively against Coulombic repulsions. The convincing characterization of the co-adsorbed adventitious hydrocarbons is indispensable with the microscopic discussion; oil vapor? At this stage, another research group cannot reproduce the experiment.

More importantly, to reveal the morphology change with co-deposition of neutral molecules and with water uptake under ambient conditions, in situ AFM measurements is inevitable because the authors discuss about the dewetting without any microscopic information on initial morphology. Depending on the coverage and the kind of halogen X for B₁₂X₁₂(²⁻), the initial morphology regarding uniformity might be changed and the image shown in Fig. S1-c) is not guaranteed experimentally.

The reviewer recommends the Editor not to accept the manuscript in Nature Communications. Since the manuscript is well written literally, it should be published in a more specialized journal relevant to applied physics after the revisions indicated below.

We thank the reviewer for making clear to us that the chemical nature of the adventitious hydrocarbons and reproducibility of the experiment were not described in sufficient detail in the original version of the manuscript. We have included comprehensive data in this context, please see point-by-point answers below.

We want to point out that the manuscript focuses on how the striking change in self-organization can be rationalized and controlled by understanding the molecular properties of the soft-landed anions. Our work reports for the first time the formation of macroscopic condensed phase liquid layers through the accumulation of mass-selected anions on a support. This bridges, in an unprecedented way, the research fields of mass spectrometry (gas phase ion chemistry) with the fields of material synthesis, thin layers and self-organization. In the revised version of the manuscript, we have rephrased the abstract and conclusions to clarify the focus of the manuscript. The discussion of the anion properties is an important and space consuming part of the manuscript. Based on the reviewer’s request, additional layer characterization has been performed and the material has been added both to the main text and supporting information. Still, not all the aspects that are of interest to the different scientific communities mentioned above are fully clarified and comprehensive follow up work that will take years is necessary. However, in the author’s opinion the manuscript has reached the limit of what a first communication (with the focus on anion properties) about this new material can cover.

[Major points]

(1) *The convincing characterization of the co-adsorbed adventitious hydrocarbons is indispensable.*

Author reply:

We have done additional experiments to identify the co-adsorbed adventitious hydrocarbons. We can clearly show that these organic molecules are predominantly phthalates with different chain length. For evidence, we have performed tandem-MS investigations on the detected organic molecules and a phthalate standard. These experiments and results are now described in section S15 of the supporting information and reference is given in the manuscript.

The referee's request is based on the concern about reproducibility. We note that the phenomena described in this study were reproduced in close to 60 soft-landing experiments carried out on our instrument over the course of 1.5 years. Recently, the instrument was taken apart and cleaned, a pump was repaired which resulted in slightly different vacuum conditions, the pump oil was replaced and seals were changed. To address the referee's concern, we reproduced soft-landing experiments with $[\text{B}_{12}\text{Cl}_{12}]^{2-}$ and $[\text{B}_{12}\text{I}_{12}]^{2-}$ under these new conditions. The reported behavior was well reproduced, e.g. $[\text{B}_{12}\text{Cl}_{12}]^{2-}$ based layers dewetted in the reported way, while $[\text{B}_{12}\text{I}_{12}]^{2-}$ layers were stable in this time frame. We further examined the composition of the adventitious hydrocarbons under these conditions. The relative abundances of the phthalates with different chain length has changed. These results confirm the experimental results are reproducible despite some measurable changes in the composition of the neutrals. We added these data to the supporting information (S15). The experiment is reproducible as long as the general nature of the hydrocarbons is preserved. In addition, we added information about vacuum conditions along with pump oil and seals used in our system to the experimental section. Therefore, in the case that another group builds a comparable instrument, the experimental conditions used in our experiments are available to them.

(2) *On initial morphology, in situ AFM measurements is inevitable to identify uniformity molecularly.*

Author reply: Unfortunately, we are not equipped to conduct *in situ* AFM experiments suggested by the reviewer. Nevertheless, we addressed this suggestion by carrying out additional *ex-situ* AFM measurements on a surface, which was stored under nitrogen to avoid dewetting prior to characterization. We provide now, in the supporting information, AFM data that show the morphology of layers that have not dewetted, see section S16. We observe that the as-deposited layers are very uniform.

[Minor points]

L24; *In abstract, the authors state "rational control of macroscopic properties", but the meaning of macroscopic properties is not clear.*

Author reply: This has been rephrased.

L95-96; *The thickness of anionic layer should be defined, corresponding to 1013, 1014 and 1015 ions. Although it is not easy to define the monolayer (ML), the amount of ions is not a good index for the thickness. Depending the deposition area size, the local thickness will be varied.*

Author reply: We have included section SI 6 that shows data in the context of the thickness of the deposited spots.

L102; In Fig. 1 b), height is changed with time?

Author reply: We apologize, we do not understand this comment.

L63; Although the chemical composition of F-SAM (1H,1H,2H,2H-perfluorodecanethiol is shown in the supplementary information in SI-14, it is better to show the chemical composition of fluorinated thiols for fluorinated self-assembled monolayer surfaces also in the text for the reader.

Author reply: We have included this on page 2.

L108-109 The authors state that “observed IR signals grew continuously during ion deposition in vacuum”. The reviewer wonders if intensity increases linearly against the deposition time without charge repulsions?

Author reply: We have included a Figure in section SI 5 that shows the almost linear increase of the signal with amount of deposited material. In section SI 1 we explain the two plate capacitor model which rationalizes low repulsion on approaching ions from already deposited ions due to the image charges in the underlying surface.

L115-116; The authors state that “We propose that adsorption of polar neutral molecules from the instrument background is driven by the presence of large numbers of ions on the surface”. Is it possible to reproduce the adsorption in another laboratory? The contamination may be native to the author’s experimental setup.

Author reply: As described in the answer to the first major comment, we carried out several additional experiments to address the question about reproducibility and provided a more detailed description of the experimental conditions.

L128; The authors state that “the layers are stabilized by co-adsorption of neutral molecules from the gas phase”. From where the neutral molecules come? Is there any evidence that the neutral molecules deposited survive as neutral species?

Author reply: Other groups have investigated the presence and origin of phthalates in the vacuum system of mass spectrometers. We added a comment on page 4 and reference 51.

The charge state of the molecules adsorbed on the surface is difficult to measure. However, the fact that they are detected as sodium adducts in positive mode ESI-MS analysis points to their original neutral charge state on the surface.

L141; “background pressure” What is main contamination in residual gas? How do they control the

background pressure?

Author reply: The background pressure is measured with an ion gauge. We added a detailed description of the vacuum system to the experimental section SI 15. The background pressure is determined by the design of the vacuum system and can only be increased by leaking neutral molecules, which was done in the experiment with glycine. The composition of the background gas may also change after the instrument is taken apart for cleaning or troubleshooting, please see answer to the first major point.

L245; Glycine vapor pressure at room temperature and at heated temperature should be shown. How much is the base pressure without glycine? The background pressure of $\sim 10^{-5}$ Torr should be compared to the base pressure without glycine. How much is the heating temperature?

Author reply: We have provided additional information about the base pressure to the experimental description and how it increased during the glycine experiment. In the case of the 70°C experiment, we did not detect a significant pressure increase, however, IR investigations clearly confirmed the codeposition during ion soft-landing and the layer showed remarkably different behavior as shown in the manuscript.

L267-275; The discussion about Figure 6 is unclear. The scientific or engineering discussion on Figure 6 should be added to clarify the focus, rather than a long figure caption of Figure 6.

Author reply: In response to this comment, the description in the main text has been significantly expanded. The Figure shows different methods that can be combined with the molecular control over self-organization to receive a high degree of control over the final meso-scale structure of the layer.

Reviewers' comments:

Reviewer #1 (Remarks to the Author):

my congratulations to the authors for their work!

Reviewer #2 (Remarks to the Author):

The revision of the paper 'self-organizing layers from complex molecular anions' presents significant additional material towards an enhanced chemical and structural characterization and analysis of the fabricated layers, while the main structure of the manuscript remains mostly the same. In particular, the claim that the layers incorporate neutral organic molecules, organic while they are in vacuum ($1e-5$ torr) and water when exposed to higher partial pressures of water in an ambient environment was made more convincing.

While the manuscript now presents a large amount of additional data, making the main finding of more convincing, the text remains descriptive and lacks to make a clear point on the origin of the observed effects. Thus, as it is now the paper is not interesting for a general readership, even though the work is relevant and novel. Below I assembled a few comment of how this could be improved, mainly focusing on changes in the style and presentation. This I would consider the minor changes needed to improve the paper to make it relevant for a general audience. Otherwise it is more likely suitable for a more specialized journal.

/1/ The introduction contains a large amount of perspective on the past development of the method while it does not make a clear point what the advancement that is made in this work. It is only said that layers were made and characterized, which is hardly novel.

/2/ Many results in the work are described in an anecdotal fashion. For instance page 8, line 256. A more systematic approach to the various observed effects and their relation would make the manuscript much more comprehensible. Most important an overarching relation is missing which can explain the findings, which would allow for a much more quantitative understanding of the involved driving forces.

For instance, it appears that the exchange in coadsorbant molecule is driven by the chemical potential. Clearly, the change in the partial pressure of water vapour can be interpreted as such: $\Delta \mu = RT \ln(p_1/p_0)$, for $p_0=1e-5$ mbar and $p_1=31$ mbar (vapour pressure of water at 25C) I approximate 20kJ/mol, a significant energy, which could explain the transition from incorporation of phthalates vs. water. See also /3/.

/3/ The incorporation of the residual phthalates seems to be favorable because it is a surfactant. Was this considered, for instance by offering other reference compounds, mediating between the hydrophobic nature of the substrate and the very polar adsorbate.

4/ The lack of water can only be directly seen in ONE IR spectrum in the supporting information, which shows the region around 3500 cm^{-1} . This is a very important point and clearly a large amount of IR spectra were measured. Certainly a comparison of an empty reference (no water), a spectrum in vacuum, a spectrum after the exposure of water and a reference with water would be the most convincing for the very relevant point.

/5/ The statement that for the first time classical methods for the analysis of such layers were used is incorrect. Ref [3] below employs XRD on soft landed molecular multilayers.

/6/ Finally, I here offer an interesting perspective on a similar experiment, which I forgot to add to my last report. [1][2]

[1] Budina, David, Julia Zakel, Johannes Martin, et al. 2014. Bombardment Induced Transport of Rb^+ through a K^+ Conducting Glass vs. K^+ Transport through a Rb^+ Conducting Glass. Zeitschrift für Physikalische Chemie. 228(4-5): 609-627.

[2] Veronika Wesp, Matthias Hermann,... Karl-Michael Weitzel: Bombardment induced ion transport – part IV: ionic conductivity of ultra-thin polyelectrolyte multilayer films. PCCP 2016, 18, 4345

[3] Rauschenbach, S.; Rinke, G.; Malinowski, N.; Weitz, R. T.; Dinnebier, R.; Thontasen, N.; Deng, Z.; Lutz, T.; de Almeida Rollo, P. M.; Costantini, G.; Harnau, L. & Kern, K.

Crystalline Inverted Membranes Grown on Surfaces by Electrospray Ion Beam Deposition in Vacuum
Adv. Mater., 2012, 24, 2761-2767

Reviewer #3 (Remarks to the Author):

The authors revised the manuscript by adding some AFM images, but one of intrinsic points is not improved sufficiently; the convincing characterization of the co-adsorbed adventitious hydrocarbons. In L130-131, the authors state that “phthalates with different chain length, which are typical plasticizers used in components of the vacuum apparatus”. The identification of the contaminations might be true. The authors should explicitly examine the exposure of phthalates toward the soft-landed anion surfaces. Otherwise, the reproducibility cannot be guaranteed in another experimental setup. Some uncontrolled contaminations are not attractive to design well-organized experiments and fabrications. Therefore, the reviewer again recommends the Editor not to accept the manuscript in Nature Communications. Since the manuscript is well written literally, it should be published in a more specialized journal relevant to applied physics after the revisions indicated below.

[minor points]

L135-138 The authors state that “these results provide strong evidence that 1) most of the deposited anions stay intact and charged on the FSAM and 2) the layers are stabilized by co-adsorption of neutral components from the gas phase background of the vacuum chamber.

When 1×10^{15} ions are accumulated into the volume of 3 mm in diameter 150 nm height, the ion density is calculated to about 20^{21} cm³. To avoid Coulombic explosions, are substantial portions compensated by counter cations resulting from neutral molecules? Some charge balance should be clarified in terms of Coulombic repulsions.

L102 (in original manuscript) L107 (in revised manuscript) ; In Fig. 1 b), height is changed with time?

The authors show the optical image in Fig. 1b). The pattern of circular holes seems to be composed of aggregates of ions. Although the width change of the pattern can be recognized, the height should also change with time. The authors should make a comment on the height change.

Supporting Information

Figure S2 Length scale should be added.

Figure S28 Length scale should be added. Is there any reason to show the image in a tilted manner?
The reviewer thinks it is better to rotate it properly.

Response to Referee comments

Referee 1

my congratulations to the authors for their work!

We appreciate the positive feedback.

Referee 2

The revision of the paper 'self-organizing layers from complex molecular anions' presents significant additional material towards an enhanced chemical and structural characterization and analysis of the fabricated layers, while the main structure of the manuscript remains mostly the same. In particular, the claim that the layers incorporate neutral organic molecules, organic while they are in vacuum ($1e-5$ torr) and water when exposed to higher partial pressures of water in an ambient environment was made more convincing.

While the manuscript now presents a large amount of additional data, making the main finding of more convincing, the text remains descriptive and lacks to make a clear point on the origin of the observed effects. Thus, as it is now the paper is not interesting for a general readership, even though the work is relevant and novel. Below I assembled a few comment of how this could be improved, mainly focusing on changes in the style and presentation. This I would consider the minor changes needed to improve the paper to make it relevant for a general audience. Otherwise it is more likely suitable for a more specialized journal.

/1/ The introduction contains a large amount of perspective on the past development of the method while it does not make a clear point what the advancement that is made in this work. It is only said that layers were made and characterized, which is hardly novel.

We revised the introduction to emphasize the novelty of the findings early on. These are: 1. It is now possible to deposit enough mass-selected ions on surfaces that condensed-phase liquid like layers form 2. the material is formed by accumulation of anions and neutrals from the gas phase and 3. these materials undergo self-organization, relevant to different applications, that may be sensitively controlled by the nature of the mass selected anions. These aspects are, in our opinion, of most interest to the general reader. In addition, we shortened the introduction by shifting some technical aspects into the results section of the manuscript. In the revised manuscript, the affected text passages are highlighted in turquoise.

The referee is right that a large amount of perspective on past development is given in the introduction. We believe that giving reference to the recent important contributions in the field of soft landing is adequate. These perspectives/reviews contain multiple references that the referee suggested us to cite in the first and second revision cycle.

/2/ Many results in the work are described in an anecdotal fashion. For instance page 8, line 256. A more systematic approach to the various observed effects and their relation would make the manuscript much more comprehensible.

To provide a more systematic description of the results, we divided the largest section with the former title “Influence of anion properties, type of surface and neutral molecules” into several new sections with the titles 1. Influence of anion properties, 2. Influence of environmental conditions, 3. Influence of surface-layer interaction, 4. Influence of accumulated molecules from the gas phase, 5. Discussion. With respect to this restructuring, some minor editorial changes to the original text, mostly at the beginning and end of the individual sections, were done (marked in yellow) so that the influence of the parameters are more clearly structured and more comprehensible to the reader.

Most important an overarching relation is missing which can explain the findings, which would allow for a much more quantitative understanding of the involved driving forces.

We have discussed comprehensively the differences between the molecular properties of the deposited anions and the relation to the differences observed in self-organization. Thus, the molecular control over the dewetting process, is the main topic of the manuscript. However, we think that the reviewer’s concern about the missing overarching relation – the dewetting process - is based on the fact that it was not made clear enough, why dewetting occurs at all. To make this more clear, we added an additional section to the supporting information that shows the interaction between phthalates (the main component of the layers accumulated from the gas phase) with different surfaces (Supplementary Note 11 and Supplementary Figure 25). Phthalates have a high contact angle on FSAM and form droplets, as was shown by the layers after the dewetting process is complete. This observation provides an additional support to our conclusion that the final morphology after dewetting may be correlated with layer-surface interactions at the interface as was already discussed. On the other hand, the formation of a smooth stable layer in the initial state is explained by the electrostatic binding of the deposited anions to the gold surface (mirror charge model) discussed in detail in Supplementary Note 1 and Supplementary Figure 1. Generally, dewetting occurs when repulsive interfacial forces overcome attractive forces, as described comprehensively in dewetting literature that we cite. Therefore, during the slow dewetting process, the competition between electrostatic binding of the layer to the surface and the repulsive interactions at the layer surface interface shifts in favor of repulsive interactions which results in dewetting. We think that this was implied but not clearly formulated and we clarified this point in the revised manuscript by adding this explanation (highlighted in purple) to the discussion. This addition not only addresses this reviewer’s concerns but also comment /3/ and remark /1/(minor points) from referee 3. Please see detailed answers.

For instance, it appears that the exchange in coadsorbant molecule is driven by the chemical potential. Clearly, the change in the partial pressure of water vapour can be interpreted as such: $\Delta \mu = RT \ln(p_1/p_0)$, for $p_0=1e-5$ mbar and $p_1=31$ mbar (vapour pressure of water at 25C) I approximate 20kJ/mol, a significant energy, which could explain the transition from incorporation of phthalates vs. water. See also /3/.

We think that it is rather intuitive that water can be accumulated in a hygroscopic material under ambient conditions while in vacuum the high vapor pressure of water leads to a preferred accumulation of less volatile molecules (phthalates), which are a major component in the mass spectrometer background. We agree that there is a strong change in chemical potential and we added a statement that the intake of water can be additionally rationalized by a change in the chemical potential

(highlighted in blue, see discussion). However, we are reluctant to discuss the energetics in more quantitative terms, because many parameters, not only the partial pressure of water, change when the surface is exposed to air.

/3/ The incorporation of the residual phthalates seems to be favorable because it is a surfactant. Was this considered, for instance by offering other reference compounds, mediating between the hydrophobic nature of the substrate and the very polar adsorbate.

The classification of the here discussed dialkyl-phthalate esters as surfactants is debatable, since there is no strong polar group in contrast to monoalkyl-phthalates which serve as surfactants. The additional section (Supplementary Note 11, Supplementary Figure 25) that we added to the Supplementary information shows that phthalates have a high contact angle on FSAM and, therefore, cannot serve as a mediator between the deposited anions and the substrate. It follows that surface interactions with both chemical components of the layer (phthalates and anions) are repulsive, which indicates that mirror charges play a major role in the formation of the initial stable layer (see purple highlighted text and Supplementary Note 1). We have already shown that other neutrals such as glycine with very different properties may serve as neutral components of the layer.

4/ The lack of water can only be directly seen in ONE IR spectrum in the supporting information, which shows the region around 3500 cm⁻¹. This is a very important point and clearly a large amount of IR spectra were measured. Certainly a comparison of an empty reference (no water), a spectrum in vacuum, a spectrum after the exposure of water and a reference with water would be the most convincing for the very relevant point.

We are currently equipped to perform *in-situ* IR measurements under vacuum. All measurements require a background spectrum to be measured prior to deposition. The dewetting process occurs under ambient conditions over an extended period of time. A faster dewetting catalyzed by introducing water vapor inside the instrument is critical with respect to the sensitive hygroscopic IR-vacuum windows. Therefore, IR measurements on an exposed surface would not be directly comparable to the data obtained under vacuum.

Our conclusion that water does not get incorporated in significant amounts into the layer during the deposition process is not based on just “one IR spectrum” that shows the absence of water. We have added quantitative XPS data on freshly prepared layers prior to dewetting and after dewetting as additional evidence in the last revision. The referee did not comment on these extensively discussed XPS data (see last response to referees). The results from this complementary characterization provide a strong support to our conclusion. In addition, we have discussed in the manuscript that exposure to water induces the dewetting process – a phenomenon that would not be understandable if the layers already contained substantial amounts of water. The “one IR spectrum” is part of an overarching picture that results from the consideration of various complementary analytical methods.

/5/ The statement that for the first time classical methods for the analysis of such layers were used is incorrect. Ref [3] below employs XRD on soft landed molecular multilayers.

The referee points to a study that was already cited in which powder XRD was performed, which requires considerably less material than our NMR investigation. However, we understand that there is no clear definition about what constitutes a “classical method”. Therefore, we changed our wording to “For the first time, THE classical molecular structure analysis method NMR was used to analyze mass selected ions collected on a substrate at the end of a mass spectrometer” – this is, to the best of our knowledge, true. The exciting opportunities that this opens for gas phase ion chemistry were acknowledged by referee 1 in the first revision cycle.

/6/ Finally, I here offer an interesting perspective on a similar experiment, which I forgot to add to my last report. [1][2]

Reference 3 has already been cited. References 1 and 2 are, in our opinion, not directly related to the context of the current work but interesting reports in the broader context of ion soft landing. They have been cited in the introduction.

[1] Budina, David, Julia Zakel, Johannes Martin, et al. 2014. Bombardment Induced Transport of Rb+ through a K+ Conducting Glass vs. K+ Transport through a Rb+ Conducting Glass. Zeitschrift für Physikalische Chemie. 228(4-5): 609-627.

[2] Veronika Wesp, Matthias Hermann,... Karl-Michael Weitzel: Bombardment induced ion transport – part IV: ionic conductivity of ultra-thin polyelectrolyte multilayer films. PCCP 2016, 18, 4345

[3] Rauschenbach, S.; Rinke, G.; Malinowski, N.; Weitz, R. T.; Dinnebier, R.; Thontasen, N.; Deng, Z.; Lutz, T.; de Almeida Rollo, P. M.; Costantini, G.; Harnau, L. & Kern, K. Crystalline Inverted Membranes Grown on Surfaces by Electrospray Ion Beam Deposition in Vacuum Adv. Mater., 2012, 24, 2761-2767

We appreciate the many constructive suggestions made by referee 2 during the revision process.

Referee 3

“The authors revised the manuscript by adding some AFM images, but one of intrinsic points is not improved sufficiently; the convincing characterization of the co-adsorbed adventitious hydrocarbons. In L130-131, the authors state that “phthalates with different chain length, which are typical plasticizers used in components of the vacuum apparatus”. The identification of the contaminations might be true. The authors should explicitly examine the exposure of phthalates toward the soft-landed anion surfaces. Otherwise, the reproducibility cannot be guaranteed in another experimental setup. Some uncontrolled contaminations are not attractive to design well-organized experiments and fabrications. Therefore, the reviewer again recommends the Editor not to accept the manuscript in Nature Communications. Since the manuscript is well written literally, it should be published in a more specialized journal relevant to applied physics after the revisions indicated below.”

We have performed the suggested experiment. The data are shown in Supplementary Figure 9 and a corresponding sentence was added to the section “Influence of accumulated molecules from the gas phase” in the manuscript (marked in red).

However, the authors feel compelled to comment on the referee’s statement: Upon request of the referee in the last revision, we identified the chemical nature of the adventitious hydrocarbons unambiguously. This was accomplished by examining their fragmentation pathways in tandem mass spectrometry (MS/MS) experiments and comparison with MS/MS of a commercially available purified phthalate standard. This approach is widely used for unambiguous identification of molecules using mass spectrometry. To address the reviewer’s concern about reproducibility, we added material that shows the reproducibility over the timeframe of 1 ½ years and also shows that the observed dependence of the dewetting process on the anion properties is not critically affected by the exact phthalate mixture composition. The referee did not comment on these data. Since these data are mostly shown in the supporting information, we can only assume that they may have been overlooked. Although the substitution by one defined phthalate is not necessary for the identification or proof of reproducibility, such experiment strengthens the evidence that we have full control over the process. Therefore we performed an experiment with a commercially available phthalate standard by introducing diisodecylphthalate liquid into our soft landing instrument and heating the reservoir to maintain an elevated partial pressure throughout the deposition. The generated layer produced by $[B_{12}Cl_{12}]^{2-}$ deposition exhibited exactly the same dewetting behavior. Chemical analysis by mass spectrometry showed the deposited anion in negative mode and only one signal with considerable intensity corresponding to $[diisodecylphthalate + Na]^+$ in positive mode. This additional data is available in Supplementary Figure 9.

[minor points]

L135-138 The authors state that “these results provide strong evidence that 1) most of the deposited anions stay intact and charged on the FSAM and 2) the layers are stabilized by co-adsorption of neutral components from the gas phase background of the vacuum chamber.

When 1×10^{15} ions are accumulated into the volume of 3 mm in diameter 150 nm height, the ion density is calculated to about 20^{21} cm³. To avoid Coulombic explosions, are substantial portions compensated by counter cations resulting from neutral molecules? Some charge balance should be clarified in terms of Coulombic repulsions.

We refer in the manuscript to the parallel plate capacitor model that is an established model in ion soft landing used to explain charge retention of deposited ions on conductive surfaces. Since the explanation was not clear to the referee, we extended Supplementary Note 1 and Supplementary Figure 1 (marked in yellow) to explain in more detail how charging of the gold surface counterbalances the charge of the deposited anions. Added explanations summarize the results of recent reviews on the topic. The importance of the model (electrostatic binding of deposited anions to the gold surface) should be clear based on the addition made in the discussion (highlighted in purple). However, we agree with the referee that due to the large amount of ions deposited in our experiments one may assume that additional effects, resulting, for example, from partial oxidation of neutral components, may contribute to stabilization of the layer. Due to the complexity of the system, not all chemical processes that may occur within these newly discovered materials have been fully characterized in this first report, which focuses on molecular control over self-organization based on anion properties. Comprehensive follow up work will be necessary to investigate these questions in detail. We added the sentence “Future studies

will focus on detailed understanding of all charge balancing processes in the formed layers both during ion deposition and ambient dewetting process, which will help exploit the full potential of these new materials” to the conclusion (marked in grey).

L102 (in original manuscript) L107 (in revised manuscript) ; In Fig. 1 b), height is changed with time? The authors show the optical image in Fig. 1b). The pattern of circular holes seems to be composed of aggregates of ions. Although the width change of the pattern can be recognized, the height should also change with time. The authors should make a comment on the height change.

We added AFM figures obtained during the dewetting process, see Supplementary Figure 18. This sequence shows the requested data on height. A reference to this data is given in the manuscript on page 4 (highlighted in green).

Supporting Information

Figure S2 Length scale should be added.

Figure S28 Length scale should be added. Is there any reason to show the image in a tilted manner? The reviewer thinks it is better to rotate it properly.

We followed both suggestions.

Reviewers' comments:

Reviewer #2 (Remarks to the Author):

The second revision of the manuscript 'Self-organizing layers from complex molecular anions' by Warneke et al. contains a number of clarifications and additions, in particular to the supporting information.

Overall, the manuscript convincingly presents a great technical achievement that is the deposition of a large amount of material by soft landing, which lead to novel behavior of the generated film. Specifically the layers show dewetting if exposed to laboratory air which was enhance when the amount of water vapour was increased. Further the dewetting was depending on the halogen in the decaborate molecule.

These films are analyzed with a multitude of methods including IR-spectrometry, mass spectrometry, (DESI), XPS, AFM, NMR and optical microscopy. A chemical analysis shows the presence of the intact molecule as well as additional compounds in particular hydrocarbon stemming from the pump oil. It was further shown that other molecules presented as vapour are incorporated and finally that the films can be structured by electron irradiation.

While I consider the technical achievement in depositing large amount of material as substantial and the observations of the behavior of the layers on surface as highly novel and interesting, the chemical analysis as well as the interpretation and discussion that follows from it appears incomplete, overly qualitative, and hence anecdotal and unconvincing. Many observation made are not analyzed quantitatively (AFM) or if they are I have to combine the data from different places in the manuscript and SI. I am aware that the observation of very small amount of complex material is a very difficult task and also that the interpretation of a novel effect like the one observed here is complicated by the fact that it cannot easily be compared to previous observations.

Therefore, in the following, I show in detail where I see the conflicts and add suggestion of how they could be addressed. As for now, I think the manuscript is a borderline case being relevant for its achievement and novel observation, but not useful for a general audience due to the convoluted chemical analysis and how the dewetting is discussed.

(i) chemical composition

The main claims about the chemical composition is that the B₁₂Cl₁₂ molecules for layers in combination with hydrocarbons from the background gas, which is at high vacuum pressure during deposition (6e-5 Torr). At this high pressure it is convincing that a significant partial pressure of pump oils is present, but also many other gases. These can be incorporated into the film.

Clearly, the authors show this for several cases, including the incorporation of water under ambient conditions, however they claim that this is not happening in vacuum. This is not convincing. The measurements of XPS ex-situ, NMR and DESI cannot be used to look for water. While the comparison of Table 1 and Supplementary Table 2 (page 4 line 170) shows that more waters is incorporated, it does not exclude that water was not present before.

Thus the absence of water in the IR spectrum remains the only one point. This very important point is however not explicitly shown and also no reference measurements of the bare substrates are presented.

To this end, also no residual gas mass spectrum is presented. From it the partial pressures can be estimated and the kinetics of the layer formation. Still, it is absolutely unconvincing why water, with its extremely high dielectric constant would not be incorporated into the layer but instead a hydrocarbon.

In particular, the vapour pressure of these pump oils is typically significantly lower than that of water and the authors show that the films do incorporate water.

How does the layer thickness show the uptake of hydrocarbons? (page 4 line 172) When I calculate the layer volume I land in good approximation at the volume of the deposited molecules (very rough approximation). Was this data quantitatively analyzed.

(ii) Discussion

The discussion of the forces involved in the behavior of the film remains vague and unconvincing. The behaviour of the film could alternatively be explained by a shift in surface energy of the material when more water is incorporated. If the layer would really still be charged, electro-dewetting should be possible (or at least be considered by the authors), which could probably even be tested in the vacuum chamber.

page 6, line 255: being hygroscopic is a property that should not depend on the partial pressure of water (or the chemical potential).

page 7, line 260-270: The interaction overall is certainly not repulsive. It is further doubtful that only the electrostatic interaction keeps the film on the surface. In this case it could be removed with a bias voltage to the gold. Clearly dispersion forces are ignored in this discussion.

Reviewer #3 (Remarks to the Author):

The authors again revised the manuscript. The revisions on the minor points pointed out by the reviewer are satisfying. However, the reviewer thinks that the convincing characterization of the co-adsorbed adventitious hydrocarbons is still lacking.

The soft-landing of selected ions of anions and cations, which is used in this work, is indeed the state of the art, because chemically unique composition can be identified through highly resolved mass-spectrometry. Recently, to bridge between gaseous ion chemistry and deposited material science, as mentioned in this manuscript, it is indispensable to enhance ion beam intensity.

At the same time, however, when the soft-landing/deposition of the targeted ions is done, the characterization of the deposited substrate should be controlled. Since it takes rather longer time to accumulate the ions sufficiently on a substrate, it is necessary to prepare the deposited substrate under ultra-high vacuum level to avoid contaminations and also to guarantee the chemical identification and uniformity. A lot of studies relevant to the soft-landing have been performed under higher vacuum level than 1×10^{-7} mbar "at least" and ideally less than 1×10^{-9} mbar. Although the dewetting process induced by ion depositions is the main topic of the manuscript, the preparation ways seem not convincing under the phthalates contaminations. Even with interesting macroscopic observation of the dewetting, the preparation and characterization of the ion deposited substrate should be also state of the art. The reviewer is still afraid that the reproducibility as well as chemical characterization cannot be guaranteed owing to some uncontrolled contaminations.

Therefore, the reviewer again recommends the Editor not to accept the manuscript in Nature Communications. Since the manuscript is well written literally, it should be published in a more specialized journal relevant to applied physics.

Reviewer #4 (Remarks to the Author):

The article is well written and the initial reviews and corrections have been addressed. This work is cutting edge in ion-landing, and is opening a brave new area of chemical constructs.

Repetition is carried out well in this application. I would request that page 3, line 89 describe better the pressure differential, perhaps even in temperature change. This may clarify the changes observed in the substrate.

Some small minor changes would be, numbering consistency (Page 4, line 172. Page 6, line 238, better describe "dry conditions").

Point by point response to referees

Response to Referee 2:

We thank referee 2 for pointing out the technical achievements and novelty of our work and acknowledge the constructive comments especially in the discussion section. In the following, we list only the critical remarks and our response to them. We changed the order of the responses and answer the most important comments first, which need a more detailed explanation.

Chemical composition of the layers

Referee comment:

The measurements of XPS ex-situ, NMR and DESI cannot be used to look for water. While the comparison of Table 1 and Supplementary Table 2 (page 4 line 170) shows that more waters is incorporated, it does not exclude that water was not present before.

Author response:

We agree that NMR and DESI-MS cannot be used to detect water in the layers. It is also correct that the comparison of Table 1 and Supplementary Table 2 points to an increase of water content under ambient conditions. In contrast to the opinion of referee 2, we propose that **XPS can be used to quantify** the amount of water in the as-prepared layer, provided that the experiment is carefully designed and executed, as it was in our studies

To address this question, we investigated a high-coverage layer that was held under nitrogen after extraction of the gold target from the soft-landing instrument prior to XPS analysis. This allowed us to perform XPS on the intact (not dewetted) layer. XPS data are quantifiable. We found an atomic **carbon** content of **59%** and **oxygen** content of **11%**. We show that the signals originate only from the deposited layer and not from the underlying fluorinated SAM, because no fluorine signal was detected in XPS. Next we quantified the amount of oxygen bound to carbon atoms based on the chemical shift of the carbon XPS peak, which is sensitive to the molecular environment. In XPS spectra, oxygen-bound carbon can be clearly distinguished from non-oxygen-bound carbon. Quantification reveals that the 59% total carbon content can be divided into **48% non-oxygen bound carbon** (CH_x groups) and **11% oxygen bound carbon**, which

completely accounts for the overall detected oxygen content of the layer (total oxygen content is 11%) and confirms that an overwhelming majority of the oxygen atoms present in the layer prior to dewetting

are bound to carbon atoms. To make the finding more clear, we reformulated the corresponding sentence in the revised manuscript (see yellow marked text in the chapter chemical analysis)

We note that both XPS (Table 1, Supplementary Figure 14) and FTIR data (Supplementary Figure 7) indicate that water is not a major component of the layer before exposure to air. The precise analysis of the small amounts of material present in the layers is limited by the sensitivity of these techniques and uncertainties of the quantitative measurements. While our data point to the fact that hydrocarbons are the dominant component accumulated in the layers, trace amount of water may be present in the layers since water is a component of the background gases in the instrument. We revised the relevant statements in the chapter chemical analysis (yellow) and the discussion (light purple) to account for this possibility. However, our experiments indicate that the small amount of water possibly present in the layer before dewetting does not initiate the dewetting process, which becomes observable only in the presence of air moisture.

Referee comment:

Thus the absence of water in the IR spectrum remains the only one point. This very important point is however not explicitly shown and also no reference measurements of the bare substrates are presented.

Author response:

The IR spectrum provides additional support for the absence of water in the layer that is also inferred from the XPS results. We show the absence of water in the IR spectrum in Supplementary Figure 7: No water band is observed above 3000 cm^{-1} . We changed the diagram so that the baseline measurement (bare substrate) before the start of deposition is now clearly visible.

Referee comment:

The main claims about the chemical composition is that the B12Cl12 molecules for layers in combination with hydrocarbons from the background gas, which is at high vacuum pressure during deposition ($6\text{e-}5$ Torr). At this high pressure it is convincing that a significant partial pressure of pump oils is present, but also many other gases. These can be incorporated into the film.

Author response:

It is correct that, in addition to hydrocarbons, other gases (for example water with even higher partial pressure) are present in the vacuum chamber at this operating pressure. However, the incorporation of molecules into the layer under vacuum at room temperature depends not only on the partial pressure of the molecules in the gas phase. At room temperature, the sticking coefficients of gaseous molecules may be much smaller than one and may differ substantially for the individual molecules. The layers form through a gradual deposition of individual anions and adsorption of individual neutral molecules from the gas phase. The retention time of molecules on the surface is strongly dependent on their binding energy to dodecaborate anions (we note that no accumulation of molecules on the surface is observed in the absence of anions). Since the hydrocarbons incorporated into the layer are mainly phthalates based on MS analysis, we added a comparative computational investigation of the binding of a water molecule and diisodecylphthalate molecule to $[\text{B}_{12}\text{Cl}_{12}]^{2-}$. (Computational level B3LYP/def2-tzvp including dispersion forces, marked purple in the discussion and supplementary note 18). The computed binding

energies are approximately 10 kcal/mol (water) and 30 kcal/mol (diisododecylphthalate). Next, we used the Arrhenius equation to estimate the average residence time of water and diisododecylphthalate at room temperature on the $[B_{12}Cl_{12}]^{2-}$ surface. The results indicate a residence time of less than 1 ms for an individual water molecule and decades for the phthalate molecules. Our investigations further revealed that the surface area of the dodecaborate accommodates two phthalate molecules efficiently chelating the $[B_{12}Cl_{12}]^{2-}$, see image below. We have discussed in the manuscript that the carbon/boron ratio determined by XPS is consistent with two phthalates per deposited dodecaborate anion, which is in excellent agreement with our computational results. The preferred incorporation of the phthalates reported here is also consistent with the recently reported exceptionally strong affinities of dodecaborates towards hydrophobic binding pockets in the presence of water (DOI: 10.1002/ange.201412485).

Supplementary Figure 32: Possible configuration for two phthalates binding to $[B_{12}Cl_{12}]^{2-}$. (Optimized geometry B3LYP/def2-tzvppd)

Referee comment (continues):

Clearly, the authors show this for several cases, including the incorporation of water under ambient conditions, however they claim that this is not happening in vacuum. This is not convincing.

Author response:

The referee agrees that the layer incorporates water under ambient conditions, which means that the material is hygroscopic. Hygroscopic materials can lose water under vacuum. Specifically, the “vacuum drying” process uses the phenomenon that water can be extracted from heat sensitive hygroscopic materials under vacuum which means that water is not sticking to such materials under vacuum although it does so under ambient conditions. We propose that this happens also in the case of our layers. We have shown that water intake drives the dewetting under ambient conditions. We added experimental results that show the effect of vacuum drying on our layers in the initial stage of dewetting. After one week in vacuum, holes which have been formed in the layers did not continue growing but the borders of the holes deliquesced showing that the driving force for the dewetting was removed under long time in vacuum (supplementary note 11, supplementary Figure 20).

Referee comment:

To this end, also no residual gas mass spectrum is presented. From it the partial pressures can be estimated and the kinetics of the layer formation.

Author response:

We have shown that the kinetics of layer formation are determined by the ion flux. Specifically, the amount of phthalates binding to the surface is determined by the number of ions on the surface: IR bands of phthalates grow proportionally with the IR bands of the ion (Supplementary Note 5 and Supplementary Figure 7). Furthermore, layers that were prepared within 3 days (lower ion flux) showed no difference in the macroscopic behavior in comparison with layers of the same ion coverage prepared in 1 day (higher ion flux). The influence of the partial pressure of the background gases on layer formation can only be important if an excess of ions was present on the surface, which is not the case even at our highest ion fluxes.

We recorded a residual gas analyzer spectrum obtained during the experiment with the defined hydrocarbon (diisodecylphthalate). This mass spectrum shows qualitatively that phthalates, water and other typical background gases are present in vacuum. We added this spectrum to the supplementary information (Supplementary Figure 26) together with a literature electron impact mass spectrum of phthalates (marked in grey in the supplementary information).

Referee comment:

Still, it is absolutely unconvincing why water, with its extremely high dielectric constant would not be incorporated into the layer but instead a hydrocarbon.

Author response:

The dielectric constant is a property of bulk phase materials. Although our layers can be considered to be condensed phase materials in the end, we build the layers over time through a gradual deposition of individual anions and simultaneous adsorption of individual neutral molecules from the vacuum background. The attraction of water to the anions is governed by the ion-dipole interaction. The dipole of water (1.82 D) is small (in contrast to its very high dielectric constant, see discussion of this effect: DOI 10.1103/PhysRevLett.98.247401) in comparison to phthalates, for which dipole moments from 3-9 D have been reported (DOI: 10.1080/10659360500320602, ISBN: 978-0-323-28659-6). Water molecules do not stick as efficiently to the layer compared to phthalates, as discussed above. Therefore a condensed phase of water does not build up on the surface in vacuum and the dielectric constant does not play a role in this process.

Referee comment:

In particular, the vapour pressure of these pump oils is typically significantly lower than that of water and the authors show that the films do incorporate water.

Author response:

Please note that the hydrocarbons have been identified to be predominantly phthalates using ESI-MS and CID. It is reasonable to assume that in our deposition system collisions of water molecules with the surface are more frequent than collisions of hydrocarbons. As discussed in detail above, since our experiments are conducted at room temperature, these water molecules leave the surface in less than 1

ms. A droplet of water dropped on a surface in our instrument directly evaporates if we pump down the system due to the relatively high vapor pressure of bulk phase water at ambient temperature. A droplet of diisodecylphthalate, in comparison, evaporates very slowly. Even after day in the vacuum of our instrument, residues of the phthalate droplets are still visible on the chamber wall. Therefore, we propose it is convincing that a substance with very low vapor pressure (phthalates) stays in the condensed phase (layer), while a substance with relatively higher vapor pressure (water) escapes the layer under vacuum at room temperature.

Layer thickness

Referee 2 wrote in the initial report which we received on Sep. 9th 2017:

“Directly related to the point of the chemical composition of the films is the question after the volume of the material observed in the different stages. While the presence of $1e15$ ions ($z=-2$) on a spot of 3mm diameter suggests a film thickness of approx. 10 nanometers, the optical images obtained show interference fringes, corresponding to significant swelling reaching the dimensions of the wavelength. So there is a significant uptake of material, which is however not quantified.”

The approximation of the thickness by the referee appears reasonable to us and is in line with our own estimations. We investigated a layer via AFM (coverage $9e14$ ions) in the initial stage of dewetting and observed a layer thickness of 55 nm, which is more than 5 times higher than the value estimated by the referee. Therefore, we agreed with the referee’s comment that the layer thickness indicates a significant uptake of material and added the following text to the manuscript to clarify this point: “showing the substantial uptake of hydrocarbons and water by the deposited anions”.

In the present report, referee 2 comments:

How does the layer thickness show the uptake of hydrocarbons? (page 4 line 172) When I calculate the layer volume I land in good approximation at the volume of the deposited molecules (very rough approximation). Was this data quantitatively analyzed.

We apologize, it appears that we misunderstood the referee’s earlier comment. Interference fringes are only observed at hole borders or droplets, which can be an order of magnitude higher in height than the layer itself before dewetting (see Supplementary Figure 4 and 5). We conclude that such thickness estimations based on the number of ions can only lead to a very rough approximation and high uncertainty in the calculated volume. We deleted “showing the substantial uptake of hydrocarbons and water by the deposited anions” (sentence is marked in yellow in the chapter “Macroscopic layer behavior”).

Structure of the manuscript

Referee comment:

Many observation made are not analyzed quantitatively (AFM) or if they are I have to combine the data from different places in the manuscript and SI.

Author response:

Multiple quantitative AFM results are presented on the $[B_{12}Cl_{12}]^{2-}$ based layer which are listed below.

- Coverage dependent layer thickness in the initial stage of dewetting of $[B_{12}Cl_{12}]^{2-}$ based layers (manuscript, Supplementary Note 16, Supplementary figure 17)
- Height and shape development of hole borders during dewetting, merging of borders and droplet formation in a $9 \cdot 10^{14}$ ions coverage $[B_{12}Cl_{12}]^{2-}$ based layer (supplementary figure 18).
- Statistical roughness determination (RMS values) of the initially formed layer and the underlying surface

Because of the space limitations, most of these results are shown in the supplementary information. In response to the referee's comment we have restructured the first part of the results section **by dividing the first chapter into three subchapters (marked in yellow in the manuscript)**.

1. **Preparation of an anion-based layer.**
2. **Macroscopic layer behavior.** (optical microscopy and AFM measurements)
3. **Chemical analysis.** (IR, DESI, XPS and NMR are successively discussed)

References to these AFM results have been combined in the manuscript (marked in yellow in the manuscript). Additional quantitative AFM data are presented later in the manuscript, for example, in the case of defined structure formation by electron beam patterning. We think that the reference to these data should be given in the corresponding chapter. **We added a description of the structure of the results section in the beginning of the chapter (yellow).** To allow the reader to search for results obtained using one experimental method, we inserted a color coded table of content at the beginning of the supplementary information sorted by experimental method. **(see supplementary information table of content)**

Discussion

Referee comment:

The discussion of the forces involved in the behavior of the film remains vague and unconvincing. The behaviour of the film could alternatively be explained by a shift in surface energy of the material when more water is incorporated.

Author response:

We agree with the underlined comment from the reviewer. We intended that the argument already be a part of our discussion but the wording of the referee is clearer than ours. In the reformulated discussion **(marked in light purple)** we reorient to the referee's formulation.

Referee comment:

If the layer would really still be charged, electro-dewetting should be possible (or at least be considered by the authors), which could probably even be tested in the vacuum chamber.

Author response:

We agree that the phenomenon of electro(de)wetting should be referenced here. Phthalates themselves – a major component of the layer - have a high contact angle on FSAM but we observe a smooth layer prior to exposure to the ambient air. We propose that the previously studied capacitor situation (Supplementary note 1) may be responsible for the attractive force between layer and substrate that compensates for the unfavorable interaction of phthalates with a surface in the absence of the anions. The formation of a smooth layer containing anions and phthalates resembles electrowetting – a well-known process, in which an electric field is used to reduce the contact angle of liquids on surfaces. **We revised the discussion section to address this point (marked in turquoise).**

It remains unclear to us what the referee suggests for us to test in the vacuum chamber. We emphasize that no dewetting occurs in vacuum.

Referee comment:

page 6, line 255: being hygroscopic is a property that should not depend on the partial pressure of water (or the chemical potential).

Author response:

We changed the wording to “**these hygroscopic layers**”

Referee comment:

page 7, line 260-270: The interaction overall is certainly not repulsive. It is further doubtful that only the electrostatic interaction keeps the film on the surface. In this case it could be removed with a bias voltage to the gold. Clearly dispersion forces are ignored in this discussion.

Author response:

We understand from the reviewer’s comment that our argument was not clearly explained. The phenomenon of dewetting arises from the interplay of unfavourable surface interactions and attractive intermolecular forces which include a variety of different components including dispersion forces. **We revised this section (turquoise) and provided additional references.**

However, we respectfully disagree with the referee that the electrostatic interaction in the layer can be easily removed by biasing the surface. As summarized in the capacitor model (Supplementary Figure 1), charging of the layer occurs as a result of anion deposition onto a thin insulating layer (SAM) on a conductive gold surface. The gold surface is connected to a potential and can be readily biased. However, this bias does not alter the field across the layer as the top of the “capacitor” cannot be connected independently to a voltage source. As a result, biasing the surface only affects the absolute values of the potentials in the system (i.e. floating the entire surface to an applied voltage) but cannot be used to change the local field across the layer.

Response to Referee 3:

Referee comment:

The authors again revised the manuscript. The revisions on the minor points pointed out by the reviewer are satisfying. However, the reviewer thinks that the convincing characterization of the co-adsorbed adventitious hydrocarbons is still lacking.

The soft-landing of selected ions of anions and cations, which is used in this work, is indeed the state of the art, because chemically unique composition can be identified through highly resolved mass-spectrometry. Recently, to bridge between gaseous ion chemistry and deposited material science, as mentioned in this manuscript, it is indispensable to enhance ion beam intensity.

At the same time, however, when the soft-landing/deposition of the targeted ions is done, the characterization of the deposited substrate should be controlled. Since it takes rather longer time to accumulate the ions sufficiently on a substrate, it is necessary to prepare the deposited substrate under ultra-high vacuum level to avoid contaminations and also to guarantee the chemical identification and uniformity. A lot of studies relevant to the soft-landing have been performed under higher vacuum level than 1×10^{-7} mbar “at least” and ideally less than 1×10^{-9} mbar. Although the dewetting process induced by ion depositions is the main topic of the manuscript, the preparation ways seem not convincing under the phthalates contaminations. Even with interesting macroscopic observation of the dewetting, the preparation and characterization of the ion deposited substrate should be also state of the art. The reviewer is still afraid that the reproducibility as well as chemical characterization cannot be guaranteed owing to some uncontrolled contaminations.

Therefore, the reviewer again recommends the Editor not to accept the manuscript in Nature Communications. Since the manuscript is well written literally, it should be published in a more specialized journal relevant to applied physics.

Author response:

The reported observations are based on the fact that organic molecules that we can control, as shown with the defined standard phthalate and the pure glycine experiments, are accumulated simultaneously with the anions throughout the soft landing process. A UHV experiment at 10^{-9} mbar with comparable coverage to that achieved in our experiments is not currently possible. Even if we developed a new UHV instrument that allowed us to combine a high flux ion beam generated by electrospray with an organic molecular beam at the surface, the introduction of prototypical “sticky” molecules like phthalates into a UHV chamber is generally considered to be an undesirable event that most UHV experimentalist try to avoid. Therefore, we have principle concerns regarding this proposed experiment. However, please note that controlled experiments using a well-defined commercially available phthalate and pure glycine were performed in a setup that operates at a base pressure of 10^{-8} Torr and has a standard operating pressure of 10^{-7} Torr (Supplementary Note 9). This operating pressure increased by one to two orders of magnitude when we introduced the purified phthalate into the deposition chamber. We described the experimental conditions for the experiment with a commercially available phthalate in greater details in supplementary note 15. (marked in grey).

Response to Referee 4:

The article is well written and the initial reviews and corrections have been addressed. This work is cutting edge in ion-landing, and is opening a brave new area of chemical constructs.

Author response:

We appreciate the positive feedback.

Repetition is carried out well in this application. I would request that page 3, line 89 describe better the pressure differential, perhaps even in temperature change. This may clarify the changes observed in the substrate.

Author response:

We added the vacuum conditions in the chapter “preparation of an anion based layer” and reformulated the corresponding sentence to make clear that the layers were brought to the ambient environment from this pressure [green]. The surface was held at room temperature during the whole process.

Some small minor changes would be, numbering consistency (Page 4, line 172. Page 6, line 238, better describe "dry conditions".

Additions to the supplementary information were made, which changes the numbers of several references. We have checked for consistency after the changes made in this revision. The dry conditions are specified now in the main text (99.99% nitrogen)

REVIEWERS' COMMENTS:

Reviewer #2 (Remarks to the Author):

First, I want to thank the authors and editor for their patience and for the spirited debate over these results and their interpretation. In the current version of the manuscript the authors present further clarification and additional data to all the points raised before.

The experimental approach seems convoluted due a mixture of experimental conditions in high vacuum and under ambient conditions, which require the multitude of additional measurements and justifications. While I do not agree with the entire string/tree of arguments, they are only secondary while the main achievement holds, which is the fabrication of the layer of anionic material and the observation of its properties. Therefore I support the publication of the manuscript.

One more note: Certainly some of the supporting information was mainly intended for the revision and hence the document might be revised and slimmed down to appropriate size to avoid confusing the readers (if the authors see fit).

Reviewer #4 (Remarks to the Author):

The changes requested were attended to, and paper is ready to publish,

Response to referees

Reviewer #2 (Remarks to the Author):

First, I want to thank the authors and editor for their patience and for the spirited debate over these results and their interpretation. In the current version of the manuscript the authors present further clarification and additional data to all the points raised before.

The experimental approach seems convoluted due a mixture of experimental conditions in high vacuum and under ambient conditions, which require the multitude of additional measurements and justifications. While I do not agree with the entire string/tree of arguments, they are only secondary while the main achievement holds, which is the fabrication of the layer of anionic material and the observation of its properties. Therefore I support the publication of the manuscript.

One more note: Certainly some of the supporting information was mainly intended for the revision and hence the document might be revised and slimmed down to appropriate size to avoid confusing the readers (if the authors see fit).

Author response:

We thank the referee for the comments. We agree that the supplementary information contains a large amount of data. However, because these data were added to address specific questions of the referees, we propose that they may also be of interest to readers who would like to see more details about the analysis. In addition, referee 4 approved publication under the condition that these data are made available. We believe that the supplementary information file is reasonably easy to navigate using the table of contents, which was introduced in the last round of revision. Therefore, we prefer to publish the supporting information in its current form.